# Choice history biases subsequent evidence accumulation

Anne E Urai[1,2‡*], Jan Willem de Gee[1,2§], Konstantinos Tsetsos[1†], Tobias H Donner[1,2,3†*]

[1]Department of Neurophysiology and Pathophysiology, University Medical Center Hamburg-Eppendorf, Hamburg, Germany; [2]Department of Psychology, University of Amsterdam, Amsterdam, Netherlands; [3]Amsterdam Brain and Cognition, University of Amsterdam, Amsterdam, Netherlands

**Abstract** Perceptual choices depend not only on the current sensory input but also on the behavioral context, such as the history of one's own choices. Yet, it remains unknown how such history signals shape the dynamics of later decision formation. In models of decision formation, it is commonly assumed that choice history shifts the starting point of accumulation toward the bound reflecting the previous choice. We here present results that challenge this idea. We fit bounded-accumulation decision models to human perceptual choice data, and estimated bias parameters that depended on observers' previous choices. Across multiple task protocols and sensory modalities, individual history biases in overt behavior were consistently explained by a history-dependent change in the evidence accumulation, rather than in its starting point. Choice history signals thus seem to bias the interpretation of current sensory input, akin to shifting endogenous attention toward (or away from) the previously selected interpretation.
DOI: https://doi.org/10.7554/eLife.46331.001

**\*For correspondence:**
anne.urai@gmail.com (AEU);
t.donner@uke.de (THD)

[†]These authors contributed equally to this work

**Present address:** [‡]Cold Spring Harbor Laboratory, Cold Spring Harbor, United States; [§]Department of Neuroscience, Baylor College of Medicine, Houston, United States

**Competing interests:** The authors declare that no competing interests exist.

## Introduction

Decisions are not isolated events, but are embedded in a sequence of choices. Choices, or their outcomes (e.g. rewards), exert a large influence on subsequent decisions (*Sutton and Barto, 1998*; *Sugrue et al., 2004*). This holds even for low-level perceptual choices (*Fernberger, 1920*; *Rabbitt and Rodgers, 1977*; *Treisman and Williams, 1984*). In most perceptual choice tasks used in the laboratory, the decision should only be based on current sensory input, the momentary 'evidence' for the decision. Thus, most work on their computational and neurophysiological mechanisms has largely focused on the transformation of sensory evidence into choice (*Shadlen and Kiani, 2013*). Yet, perceptual decisions are strongly influenced by experimental history: whether or not previous choices led to positive outcomes (*Rabbitt and Rodgers, 1977*; *Dutilh et al., 2012*), the confidence in them (*Desender et al., 2018*), and the content of the previous choice (i.e. which stimulus category was selected; *Akaishi et al., 2014*; *Fründ et al., 2014*; *Urai et al., 2017*). The latter type of sequential effect, which we call 'choice history bias', refers to the selective tendency to repeat (or alternate) previous choices. It is distinct and dissociable from effects of reward, performance feedback or subjective error awareness in previous trials.

Choice history biases are prevalent in human (*Fründ et al., 2014*; *Urai et al., 2017*), monkey (*Gold et al., 2008*) and rodent (*Busse et al., 2011*; *Odoemene et al., 2018*) perceptual decision-making. Remarkably, this holds even for environments lacking any correlations between stimuli presented on successive trials – the standard in psychophysical laboratory experiments. Choice history biases vary substantially across individuals (*Abrahamyan et al., 2016*; *Urai et al., 2017*). Neural signals reflecting previous choices have been found across the sensorimotor pathways of the cerebral cortex, from sensory to associative and motor regions (*Gold et al., 2008*; *de Lange et al., 2013*;

*Akaishi et al., 2014*; *Pape and Siegel, 2016*; *Purcell and Kiani, 2016a*; *St John-Saaltink et al., 2016*; *Thura et al., 2017*; *Hwang et al., 2017*; *Scott et al., 2017*).

By which mechanism are choice history signals incorporated into the formation of a decision? Current models of perceptual decision-making posit the temporal accumulation of sensory evidence, resulting in an internal decision variable that grows with time (*Bogacz et al., 2006*; *Gold and Shadlen, 2007*; *Ratcliff and McKoon, 2008*; *Brody and Hanks, 2016*). When this decision variable reaches one of two decision bounds, a choice is made and the corresponding motor response is initiated. In this framework, a bias can arise in two ways: (i) by shifting the starting point of accumulation toward one of the bounds or (ii) by selectively changing the rate at which evidence for one versus the other choice alternative is accumulated. *Figure 1* illustrates these two biasing mechanisms for a simple and widely used form of accumulation-to-bound model: the drift diffusion model (DDM). Similar principles apply to more complex accumulation-to-bound models. The starting point shift can be thought of as adding an offset to the perceptual interpretation of the current sensory evidence. By contrast, the evidence accumulation bias corresponds to biasing that perceptual interpretation toward one of the two stimulus categories.

It is currently unknown which of those two principal mechanisms accounts for the choice history biases observed in overt behavior. Previous theoretical accounts have postulated a shift in the starting point of the decision variables toward the bound of the previous choice (*Yu and Cohen, 2008*; *Zhang et al., 2014*; *Glaze et al., 2015*). This is based on the assumption that the representation of the decision variable decays slowly, leaving a trace of the observer's choice in the next trial (*Cho et al., 2002*; *Gao et al., 2009*; *Gao et al., 2009*; *Bonaiuto et al., 2016*). However, choice history biases might also originate from a slower (i.e. tens of seconds) across-trial accumulation of internal decision variables – analogous to the accumulation of external outcomes in value-based decisions (*Sutton and Barto, 1998*; *Sugrue et al., 2004*). Previous experimental work on perceptual choice history biases either did not analyze the within-trial decision dynamics (*Busse et al., 2011*; *de Lange et al., 2013*; *Akaishi et al., 2014*; *Fründ et al., 2014*; *Urai et al., 2017*; *Braun et al., 2018*), or only tested for starting point biases, not accumulation biases (*Cho et al., 2002*; *Gold et al., 2008*; *Yu and Cohen, 2008*; *Gao et al., 2009*; *Wilder et al., 2009*; *Bode et al., 2012*; *Jones et al., 2013*; *Zhang et al., 2014*).

Here, we untangle how history-dependent changes in evidence accumulation and starting point contribute to history biases in overt choice behavior. Across a range of perceptual choice tasks, we

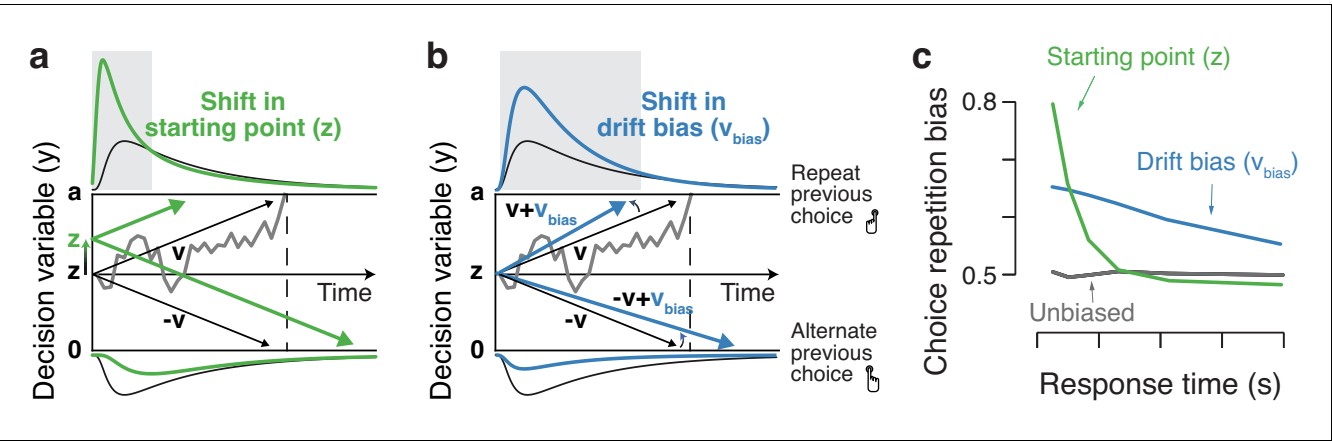

**Figure 1.** Two biasing mechanisms within the DDM. The DDM postulates that noisy sensory evidence is accumulated over time, until the resulting decision variable y reaches one of two bounds (dashed black lines at y = 0 and y = a) for the two choice options. Repeating this process over many trials yields RT distributions for both choices (plotted above and below the bounds). Gray line: example trajectory of the decision variable from a single trial. Black lines: mean drift and resulting RT distributions under unbiased conditions. (a) Choice history-dependent shift in starting point. Green lines: mean drift and RT distributions under biased starting point. Gray-shaded area indicates those RTs for which starting point leads to choice bias. (b) Choice history-dependent shift in drift bias. Blue lines: mean drift and RT distributions under biased drift. Gray shaded area indicates those RTs for which drift bias leads to choice bias. (c) Both mechanisms differentially affect the shape of RT distributions. Conditional bias functions (*White and Poldrack, 2014*), showing the fraction of biased choices as a function of RT, demonstrate the differential effect of starting point and drift bias shift.
DOI: https://doi.org/10.7554/eLife.46331.002

found that individual differences in choice repetition are explained by history-dependent biases in accumulation, not starting point. Thus, the interaction between choice history and decision formation seems to be more complex than previously thought: choices may bias later evidence accumulation processes towards (or away from) the previous chosen perceptual interpretation of the sensory input.

## Results

We fit different bounded-accumulation models to human behavioral data:choices and response times (RT). The DDM estimates model parameters from joint choices and RT distributions, and provides good fits to behavioral data from a large array of two-choice task (*Ratcliff and McKoon, 2008*). We estimated the following parameters: non-decision time (the time needed for sensory encoding and response execution), starting point of the decision variable, separation of the decision bounds, mean drift rate, and a stimulus-independent constant added to the mean drift. We refer to the latter parameter (termed 'drift criterion' by *Ratcliff and McKoon, 2008*) as 'drift bias'.

Within the DDM, choice behavior can be selectively biased toward repetition or alternation by two mechanisms: shifting the starting point, or biasing the drift toward (or away from) the bound for the previous choice (*Figure 1*). These biasing mechanisms are hard to differentiate based on the proportion of choices alone, but they are readily distinguishable by the relationship between choice bias and RT (*Figure 1c*). Specifically, the conditional bias function (*White and Poldrack, 2014*) shows the fraction of choice repetitions as a function of their RT (binned in quantiles). A shift in starting point is most influential early in the decision process: it affects the leading edge of the RT distribution and shifts its mode. It predicts that the majority of history-dependent choice biases occur on trials with fast RTs (*Figure 1c*, green). A drift bias is instead accumulated along with the evidence and therefore grows as a function of elapsed time. Thus, drift bias strongly affects the trailing edge of the RT distribution with only a minor effect on the mode, altering choice fractions across the whole range of RTs (*Figure 1c*, blue). History-dependent changes in bound separation or mean drift rate may also occur, but they can only change overall RT and accuracy: those changes are by themselves not sufficient to bias the accumulation process toward one or the other bound, and thus toward choice repetition or alternation (see *Figure 4—figure supplement 1*).

We fit different variants of the DDM (*Figure 3—figure supplement 1*) to data from six experiments. These covered a range of task protocols and sensory modalities commonly used in studies of perceptual decision-making (*Figure 2a*): two alternative forced-choice, two interval forced-choice, and yes-no (simple forced choice) tasks; RT and so-called fixed duration tasks; visual motion direction and coherence discrimination, visual contrast and auditory detection; and experiments with and without single-trial performance feedback. As found in previous work (*Fründ et al., 2014*; *Abrahamyan et al., 2016*; *Urai et al., 2017*), observers exhibited a wide range of idiosyncratic choice history biases across all experiments (*Figure 2b,c*). To ensure that the DDM is an appropriate (simplified) modeling framework for these data, we first fit a basic version of the DDM that contained the above-described parameters, without allowing bias parameters to vary with choice history. We then fit the DDM while also allowing starting point, drift bias, or both to vary as a function of the observer's choice on the previous trial.

The DDM fits matched several aspects of the behavioral data (*Figure 3—figure supplement 1*). First, RT distributions matched the model predictions reasonably well (shown separately for each combination of stimuli and choices in *Figure 3—figure supplement 1*, darker colors indicate predicted RTs obtained through model simulations). Second, for the fits obtained with a hierarchical Bayesian fitting procedure (see *Figure 3—figure supplement 1* and Materials and methods), used for *Figures 3–5*, the $\hat{R}$ for group-level parameters ranged between 0.9997 and 1.0406 across datasets, indicating good convergence of the sampling procedure (*Wiecki et al., 2013*). Third, individual drift rate estimates correlated with individual perceptual sensitivity (d', *Figure 3—figure supplement 1a*) and monotonically increased with stronger sensory evidence (*Figure 3—figure supplement 1a*). In fixed duration tasks, the decision-maker does not need to set a bound for terminating the decision (*Bogacz et al., 2006*), so the bounded diffusion process described by the DDM might seem inappropriate. Yet, the success of the DDM in fitting these data was consistent with previous work (e.g. *Ratcliff, 2006*; *Bode et al., 2012*; *Jahfari et al., 2012*) and might have reflected the fact

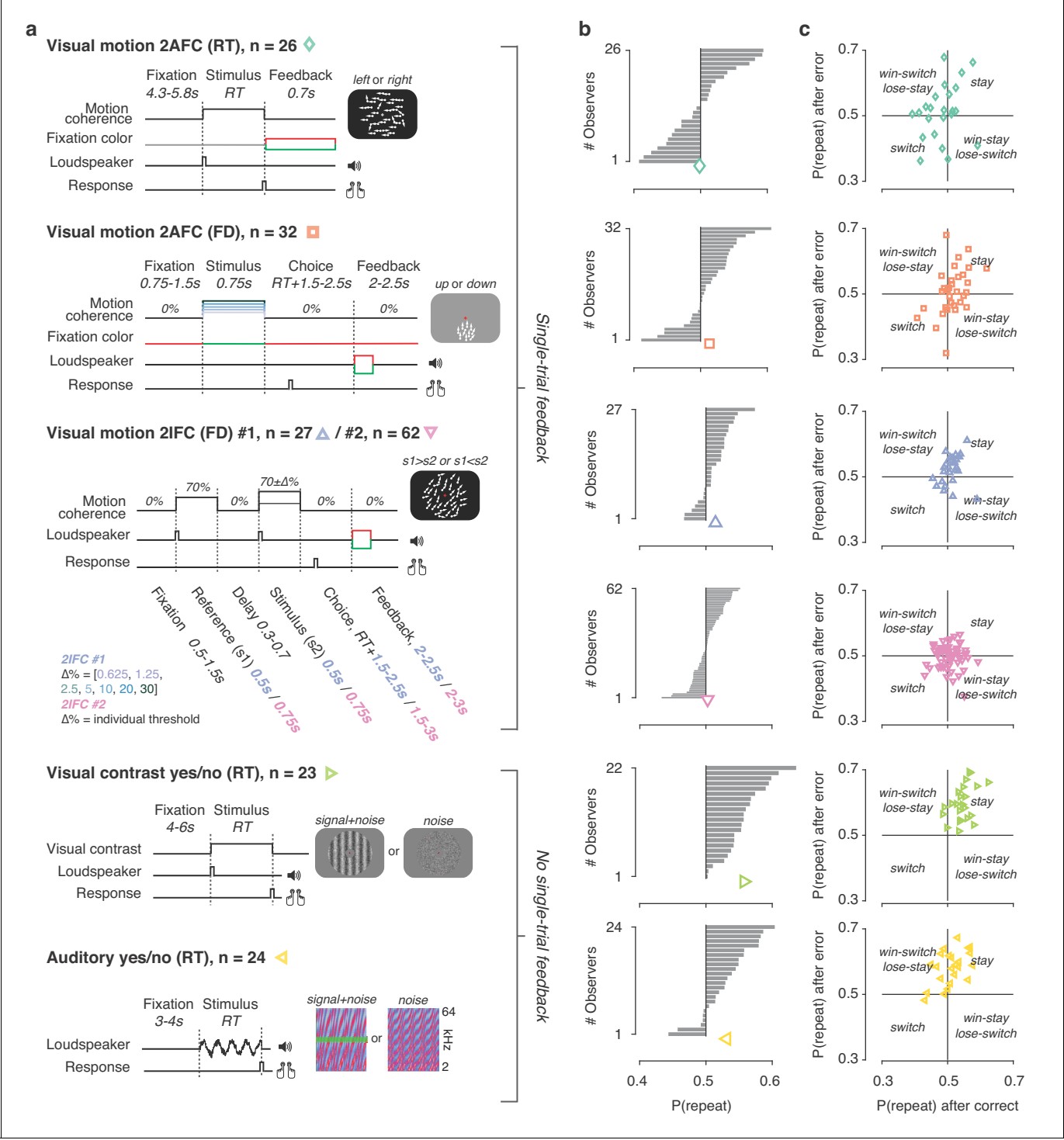

**Figure 2.** Behavioral tasks and individual differences. (a) Schematics of perceptual decision-making tasks used in each dataset. See also Materials and methods section Datasets: behavioral tasks and participants. (b) Distributions of individual choice history biases for each dataset. Gray bars show individual observers, with colored markers indicating the group mean. (c) Each individual's tendency to repeat their choices after correct vs. error trials. The position of each observer in this space reflects their choice- and outcome-dependent behavioral strategy.
DOI: https://doi.org/10.7554/eLife.46331.003

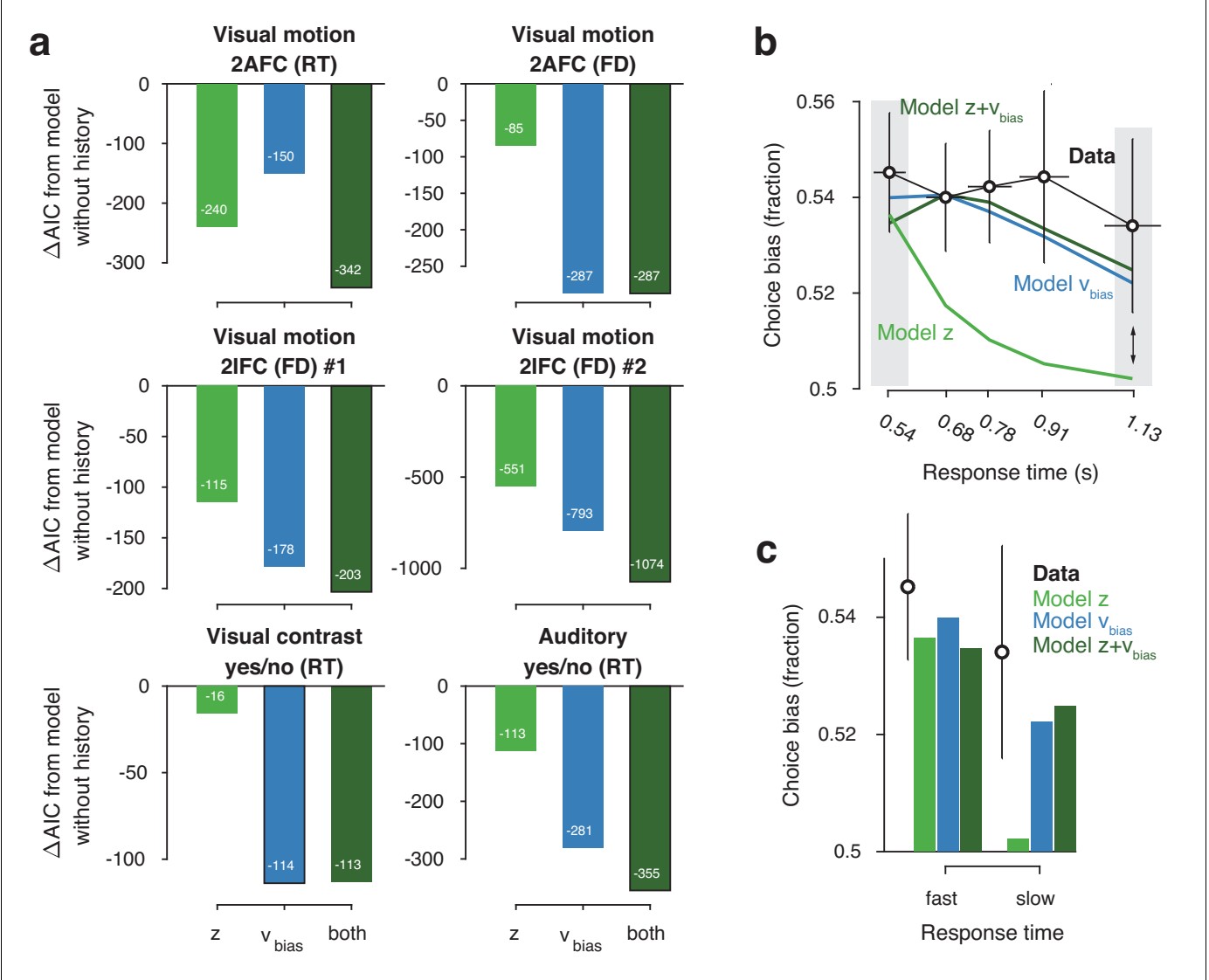

**Figure 3.** Model comparison and simulations. (a) For each dataset, we compared the AIC between models where drift bias, starting point bias or both were allowed to vary as a function of previous choice. The AIC for a model without history dependence was used as a baseline for each dataset. Lower AIC values indicate a model that is better able to explain the data, taking into account the model complexity; a ΔAIC of 10 is generally taken as a threshold for considering one model a sufficiently better fit. (b) Conditional bias functions (*Figure 1c*; *White and Poldrack, 2014*). For the history-dependent starting point, drift bias and hybrid models, as well as the observed data, we divided all trials into five quantiles of the RT distribution. Within each quantile, the fraction of choices in the direction of an individual's history bias (repetition or alternation) indicates the degree of choice history bias. Error bars indicate mean ± s.e.m. across datasets. (c) Choice bias on slow response trials can be captured only by models that include history-dependent drift bias. Black error bars indicate mean ± s.e.m. across datasets, bars indicate the predicted fraction of choices in the first and last RT quantiles.

DOI: https://doi.org/10.7554/eLife.46331.004

The following figure supplements are available for figure 3:

**Figure supplement 1.** The hierarchical DDM.
DOI: https://doi.org/10.7554/eLife.46331.005
**Figure supplement 2.** Drift diffusion model fits.
DOI: https://doi.org/10.7554/eLife.46331.006

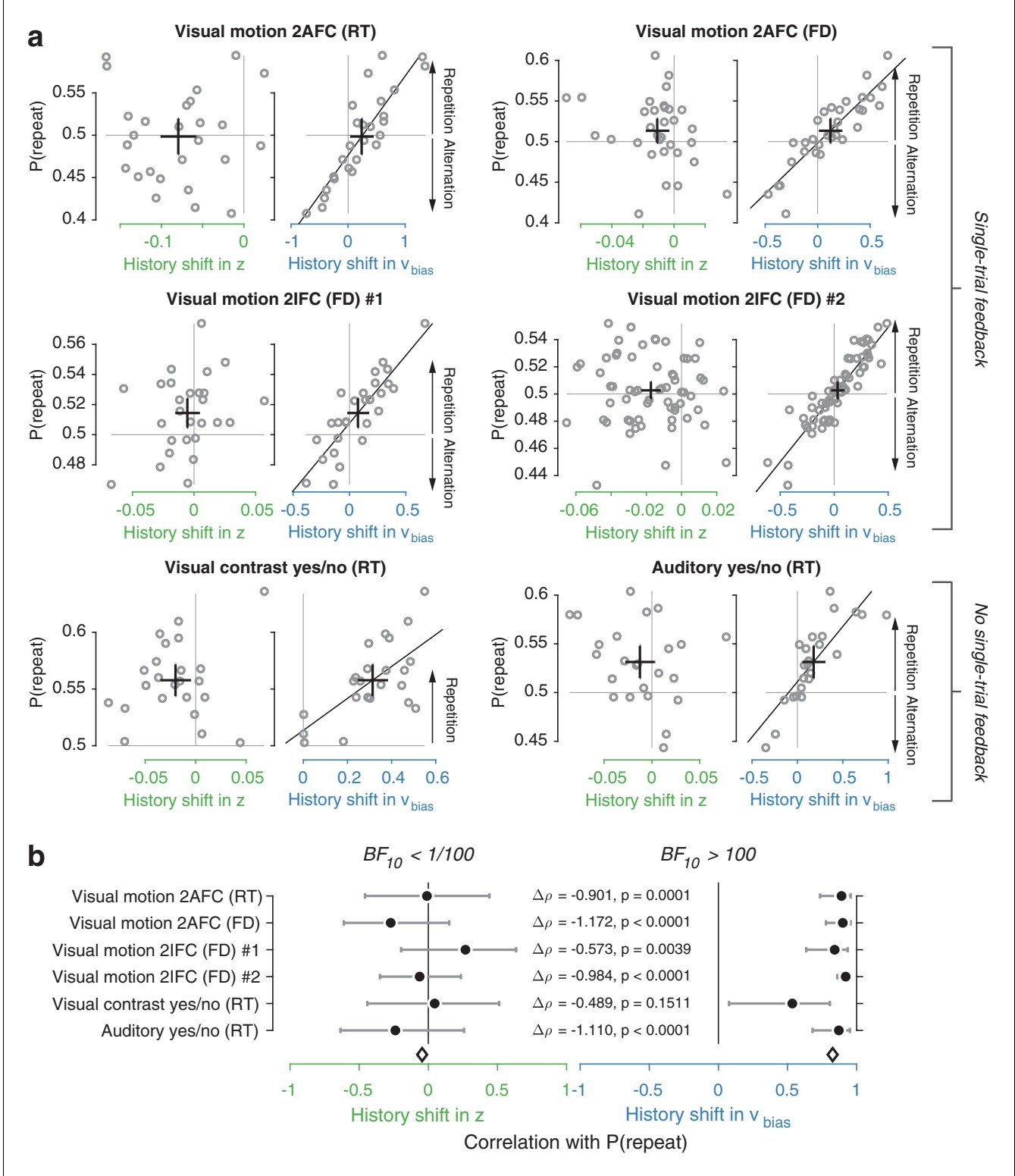

**Figure 4.** Individual choice history biases are explained by history-dependent changes in drift bias, not starting point. (a) Relationship between individual choice repetition probabilities, P(repeat), and history shift in starting point (left column, green) and drift bias (right column, blue). Parameter estimates were obtained from a model in which both bias terms were allowed to vary with previous choice. Horizontal and vertical lines, unbiased references. Thick black crosses, group mean ± s.e.m. in both directions. Black lines: best fit of an orthogonal regression (only plotted for correlations significant at p<0.05). (b) Summary of the correlations (Spearman's ρ) between individual choice repetition probability and the history shifts in starting point (green; left) and drift bias (blue; right). Error bars indicate the 95% confidence interval of the correlation coefficient. Δρ quantifies the degree to

*Figure 4 continued on next page*

*Figure 4 continued*

which the two DDM parameters are differentially able to predict individual choice repetition (p-values from Steiger's test). The black diamond indicates the mean correlation coefficient across datasets. The Bayes factor ($BF_{10}$) quantifies the relative evidence for the alternative over the null hypothesis, with values < 1 indicating evidence for the null hypothesis of no correlation, and >1 indicating evidence for a correlation.

DOI: https://doi.org/10.7554/eLife.46331.007

The following figure supplements are available for figure 4:

**Figure supplement 1.** Post-error slowing.

DOI: https://doi.org/10.7554/eLife.46331.011

**Figure supplement 2.** Control model fits.

DOI: https://doi.org/10.7554/eLife.46331.008

**Figure supplement 3.** Same biasing mechanism under two pharmacological interventions.

DOI: https://doi.org/10.7554/eLife.46331.009

**Figure supplement 4.** Repeaters vs. alternators.

DOI: https://doi.org/10.7554/eLife.46331.010

**Figure supplement 5.** Group-level posterior distributions of history bias parameters.

DOI: https://doi.org/10.7554/eLife.46331.012

that observers set implicit decision bounds also when they do not control the stimulus duration (*Kiani et al., 2008*; but see *Tsetsos et al., 2015*).

## History-dependent accumulation bias, not starting point bias, explains individual differences in choice repetition behavior

Models with history-dependent biases better explained the data than the baseline model without such history dependence (*Figure 3a*), corroborating the observation that observers' behavior showed considerable dependence on previous choices (*Figure 2f*). The model with both history-dependent starting point and drift bias provided the best fit to five out of six datasets (*Figure 3a*), based on the Akaike Information Criterion (AIC; *Akaike, 1974* - note that we obtained the same results when instead using the hierarchical Deviance Information Criterion).

The above model comparison pointed to the importance of including a history-dependency into the model. We further examined the ability of each model to explain specific diagnostic features in the data (*Palminteri et al., 2017*) that distinguished starting point from drift bias. A history-dependent shift in the starting point leads to biased choices primarily when responses are fast (early RT quantiles), whereas a history-dependent shift in drift leads to biased choices across all trials, including those with slow responses (*Figure 1*). We simulated choices and RTs from the three different model variants and computed so-called 'conditional bias functions' (*White and Poldrack, 2014*): the fraction of choices in line with each observer's choice repetition tendency (i.e. repetition probability), in each quantile of their RT distribution. For observers whose choice repetition probability was >0.5, this was the fraction of repetitions; for the other observers, this was the fraction of alternations. Consistent with a shift in drift bias, observers exhibited history-dependent choice biases across the entire range of RTs (*Figure 3b*). In particular, the biased choices on slow RTs could only be captured by models that included a history-dependent shift in drift bias (*Figure 3c*, blue and dark green bars).

We used the parameter estimates obtained from the full model (with both history-dependent starting point and drift bias) to investigate how history-dependent variations in starting point and drift bias related to each individual's tendency to repeat their previous choices. We call each bias parameter's dependence on the previous choice its 'history shift'. For instance, in the left vs. right motion discrimination task, the history shift in starting point was computed as the difference between the starting point estimate for previous 'left' and previous 'right' choices, irrespective of the category of the current stimulus. The history shift in drift bias, but not the history shift in starting point, was robustly correlated to the individual probability of choice repetition (*Figure 4a*, significant correlations indicated with solid regression lines). In five out of six datasets, the correlation with the history shift in drift bias was significantly stronger than the correlation with the history shift in starting point (*Figure 4b*, $\Delta\rho$ values).

We quantified the total evidence by computing a Bayes factor for each correlation (*Wetzels and Wagenmakers, 2012*), and multiplying these across datasets (*Scheibehenne et al., 2016*). This

further confirmed that individual choice history biases were not captured by history shifts in starting point, but consistently captured by history shifts in drift (*Figure 4b*). Specifically, the Bayes factor for the history shift in starting point approached zero, indicating strong evidence for the null hypothesis of no correlation. The Bayes factor for the history shift in drift indicated strong evidence for a correlation (*Kass and Raftery, 1995*).

Correlations between estimated history shifts in starting point and drift bias were generally negative (mean $\rho$: $-0.2884$, range $-0.4130$ to $-0.0757$), but reached statistical significance ($p<0.05$) in only one dataset. The combined Bayes Factor ($BF_{10}$) was 0.0473, indicating strong evidence for $H_0$. We thus remain agnostic about the relationship between the history shifts of both parameters.

The same qualitative pattern of results was obtained with an alternative fitting procedure (non-hierarchical $G^2$ optimization, *Figure 4—figure supplement 2a*), as well as a model that allowed for additional across-trial variability in non-decision time (*Figure 4—figure supplement 2b*). Letting non-decision time vary with each level of sensory evidence strength (in the two datasets including multiple such levels) did not change the pattern of model comparison and correlation results (*Figure 4—figure supplement 2c*). These findings are thus robust to specifics of the model and fitting method. The Visual motion 2IFC #2 also included pharmacological interventions in two sub-groups of participants (see Materials and methods); we found the same effects for both drug groups as well as the placebo group (*Figure 4—figure supplement 3*). A significant positive correlation between history shift in drift bias and P(repeat) was present for two sub-groups of participants, defined as 'repeaters' and 'alternators' (based on P(repeat) being larger or smaller than 0.5, respectively; *Figure 4—figure supplement 4*).

The lack of a correlation between history-dependent starting point shifts and individual choice repetition is surprising in light of previous accounts (*Yu and Cohen, 2008*; *Gao et al., 2009*). History shifts in starting point were mostly negative (a tendency toward choice alternation) across participants, regardless of their individual tendency toward choice repetition or alternation (*Figure 4—figure supplement 5*, significant in two out of six datasets). This small but consistent effect likely explains why our formal model comparison favored a model with both history-dependent drift and starting point over one with only drift bias (see also Discussion). Critically, only the history-dependent shift in drift bias accounted for individual differences in choice repetition (*Figure 4*).

## History-dependent accumulation bias explains individual choice repetition behavior irrespective of previous choice outcome

In four out of six tasks, participants received explicit outcome feedback (correct, error) after each choice. It is possible that participants experienced positive feedback as rewarding and (erroneously) assumed that a rewarded choice is more likely to be rewarded on the next trial. Manipulations of reward (probability or magnitude) have been found to change starting point (*Voss et al., 2008*; *Leite and Ratcliff, 2011*; *Mulder et al., 2012*), but might also bias drift (*Liston and Stone, 2008*; *Afacan-Seref et al., 2018*; *Fan et al., 2018*). Given that there were far more correct (i.e. rewarded) choices than errors, the history-dependent drift bias could reflect the expectation of reward for the choice that was correct on the previous trial.

Two findings refute this idea. First, the same result holds in the two datasets without single-trial outcome feedback (*Figure 4a*, bottom row), implying that external feedback is not necessary for history shifts in drift bias. Second, we found similar results when separately estimating the model parameters (history shift in starting point and drift bias) and model-free measures (choice repetition probability) after both correct and error trials (*Figure 5a*). Across datasets, individual repetition probability was best explained by history shifts in drift bias, not starting point, after both correct (*Figure 5b*) and error (*Figure 5c*) trials. Thus, even erroneous choices bias evidence accumulation on the next trial, in the same direction as correct choices. Indeed, most participants were predominantly biased by their previous choice (95 'stay', 30 'switch'), while a third was biased by a combination of the previous choice and its correctness (26 'win-stay lose-switch', 42 'win-switch lose-stay'; *Figure 2c*).

Correlations tended to be smaller for previous erroneous choices. However, directly comparing the correlation coefficients between post-correct and post-error trials (after subsampling the former to ensure equal trial numbers per participant) did not allow us to refute nor confirm a difference (*Figure 5d*). In sum, history-dependent drift biases did not require external feedback about choice outcome and were predominantly induced by the previous choice. These choice history-dependent

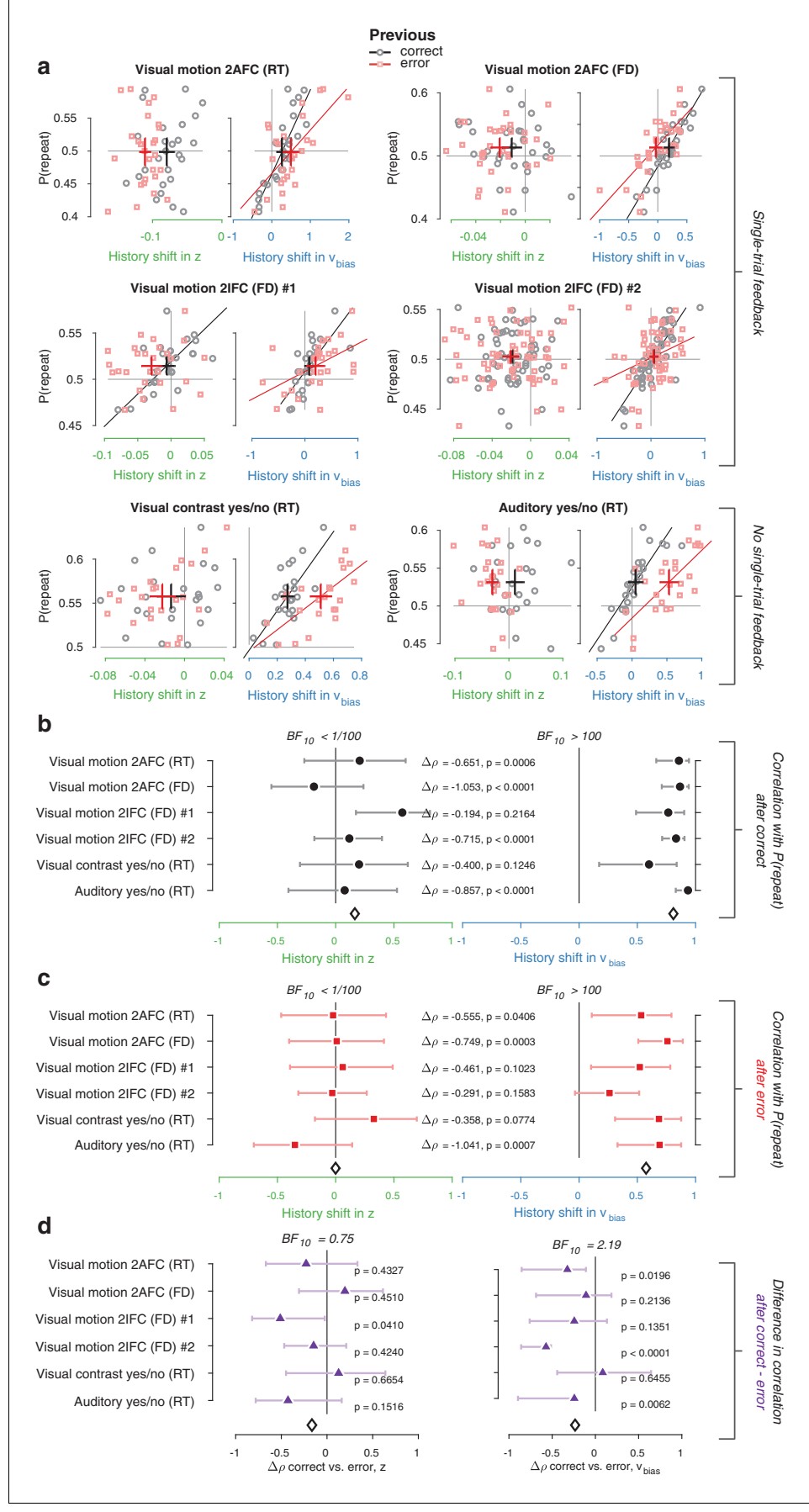

**Figure 5.** History shift in drift bias explains individual choice behavior after both error and correct decisions. As in *Figure 4*, but separately following correct (black) and error (red) trials. Post-correct trials were randomly subsampled to match the trial numbers of post-error trials. (a) Relationship between repetition probability and history shifts in starting point and drift bias, separately computed for trials following correct (black circles) and error (red squares) responses. (b) Summary of correlations (as in *Figure 4c*) for trials following a correct response. Error bars indicate the 95% confidence interval of the correlation coefficient. (c) Summary of correlations (as in *Figure 4c*) for trials following an error response. (d) Difference in correlation coefficient between post-correct and post-error trials, per dataset and parameter. Δρ quantifies the degree to which the two DDM parameters are differentially able to predict individual choice repetition (p-values from Steiger's test). The black diamond indicates the mean correlation coefficient across datasets. The Bayes factor (BF$_{10}$) quantifies the relative evidence for the alternative over the null hypothesis, with values < 1 indicating evidence for the null hypothesis of no correlation, and >1 indicating evidence for a correlation.

DOI: https://doi.org/10.7554/eLife.46331.013

biases in evidence accumulation were accompanied by effects on drift rate and boundary separation (*Figure 4—figure supplement 1*), in line with previous work on post-error slowing (*Dutilh et al., 2012*; *Goldfarb et al., 2012*; *Purcell and Kiani, 2016a*).

## Accumulation bias correlates with several past choices

Does the history shift in evidence accumulation depend on events from one past trial only? Recent work has exposed long-lasting choice history biases that span several trials and tens of seconds (*Urai et al., 2017*; *Braun et al., 2018*; *Hermoso-Mendizabal et al., 2018*). We thus estimated the influence of past events on the evidence accumulation process in a more comprehensive fashion. We fit a family of models in which correct and incorrect choices from up to six previous trials were used as predictors, and estimated their contribution to current starting point and drift bias.

Inclusion of further lags improved the model's ability to account for the data, up to a lag of 2–4 after which model fits (ΔAIC) began to deteriorate (*Figure 6—figure supplement 1*). In 4/6 datasets, the best-fitting model contained only history-dependent changes in drift, not starting point, over a scale of the previous 2–4 trials. In the other two datasets, the best-fitting model was a hybrid where both drift and starting point varied as a function of choice history, up to two to trials into the past (*Figure 6—figure supplement 1*). We computed 'history kernels' across the different lags, separately for starting point and drift bias. These are analogous to the kernels obtained from a history-dependent regression analysis of the psychometric function that ignores decision time (*Fründ et al., 2014*), and which have been widely used in the recent literature on choice history biases (*Fründ et al., 2014*; *Urai et al., 2017*; *Braun et al., 2018*). To interpret these group-level kernels in light of substantial individual variability, we expressed each regression weight with respect to individual repetition probability at lag 1 (i.e. switching the sign for alternators).

Previous choices shifted drift bias in line with individual history bias across several trials, whereas starting point did not consistently shift in the direction of history bias. The hybrid models showed that the effect of choice history on drift bias decayed over approximately three past trials (*Figure 6a*), with a slower decay than for starting point (*Figure 6a*). The regression weights for past trials (from lag two through each dataset's best-fitting lag) for drift bias – but not starting point - significantly correlated with the probability of repeating past choices at these same lags (*Figure 6b*). This was true after both correct and error trials (*Figure 6b*), similarly to the effects at lag 1 (*Figure 5b–c*).

In sum, the biasing effect of choice history on evidence accumulation is long-lasting (longer than the effects on starting point), dependent on preceding choices several trials into the past, but independent of their correctness. This analysis corroborates the previous findings from our simpler models focusing on only the preceding trial, and further dissociate the effects of choice history on starting point and evidence accumulation.

## History-dependent accumulation bias explains individual choice repetition behavior irrespective of specifics of bounded-accumulation models

We next set out to test the generality of our conclusions and gain deeper mechanistic insight into the nature of the dynamic (i.e. time-increasing) bias. We used a variety of bounded-accumulation models with more complex dynamics than the standard DDM. We focused on the preceding trial only, which our previous analyses had identified as exerting the same effect on history bias as the

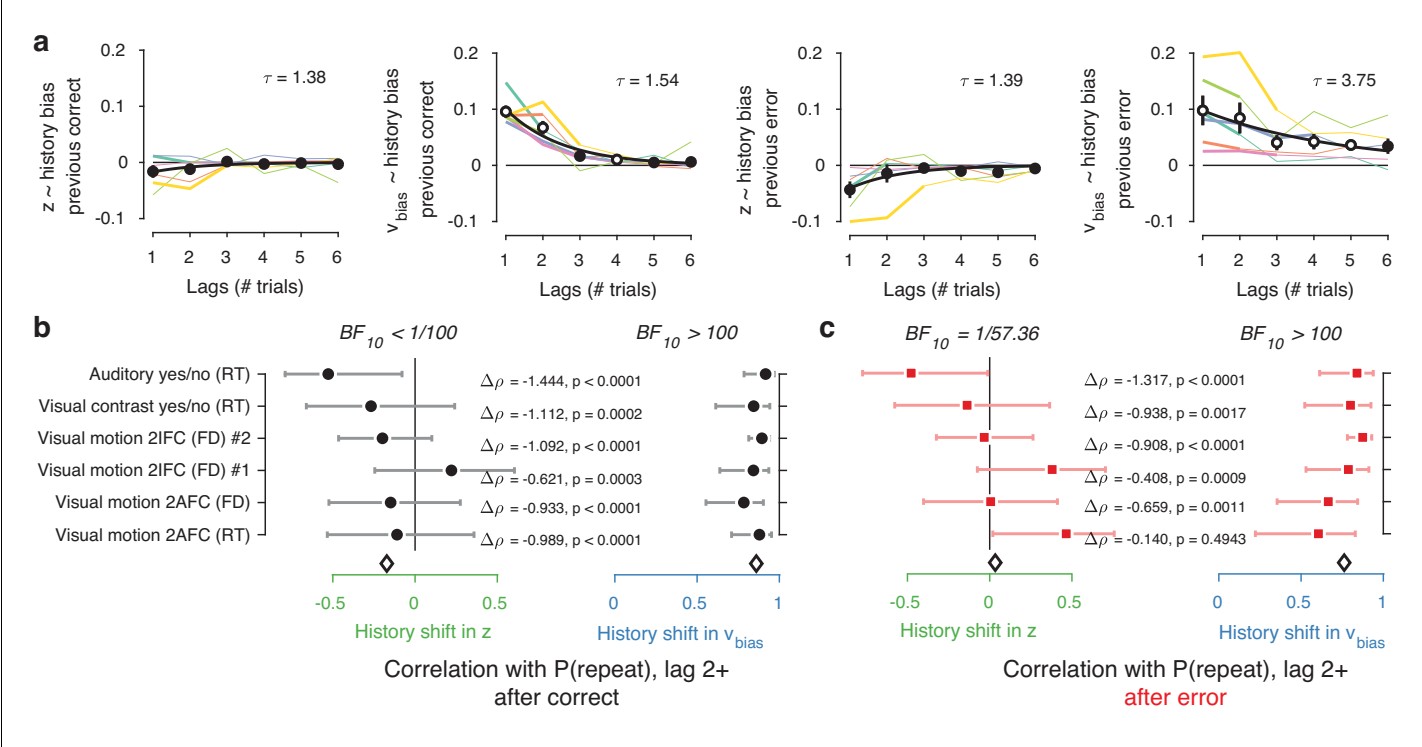

**Figure 6.** Choice history affects drift bias over multiple trials. (**a**) History kernels, indicating different parameters' tendency to go in the direction of each individual's history bias (i.e. sign-flipping the parameter estimates for observers with P(repeat)<0.5). For each dataset, regression weights from the best-fitting model (lowest AIC, *Figure 6—figure supplement 1*) are shown in thicker lines; thin lines show the weights from the largest model we fit. Black errorbars show the mean ± s.e.m. across models, with white markers indicating timepoints at which the weights are significantly different from zero across datasets (p<0.05, FDR corrected). Black lines show an exponential fit $V(t) = Ae^{-t/\tau}$ to the average. (**b**) Correlations between individual P(repeat) and regression weights, as in *Figure 5b–c*. Regression weights for the history shift in starting point and drift bias were averaged from lag two until each dataset's best-fitting lag. P(repeat) was corrected for expected repetition at longer lags given individual repetition, and averaged from lag two to each dataset's best-fitting lag. Δρ quantifies the degree to which the two DDM parameters are differentially able to predict individual choice repetition (p-values from Steiger's test). The black diamond indicates the mean correlation coefficient across datasets. The Bayes factor (BF$_{10}$) quantifies the relative evidence for the alternative over the null hypothesis, with values < 1 indicating evidence for the null hypothesis of no correlation, and >1 indicating evidence for a correlation.

DOI: https://doi.org/10.7554/eLife.46331.014

The following figure supplement is available for figure 6:

**Figure supplement 1.** Contribution of previous choices to current drift and starting point bias as function of lag.

DOI: https://doi.org/10.7554/eLife.46331.015

longer lags (*Figure 6*). These models included variants of the DDM (i.e. a perfect accumulator) with more complex dynamics of the bias or the decision bounds, as well as variants of a leaky accumulator (*Busemeyer and Townsend, 1993*; *Usher and McClelland, 2001*; *Brunton et al., 2013*). We focused on the Visual motion 2AFC (FD) dataset because it entailed small random dot stimuli (diameter 5° of visual angle), leading to large within- and across-trial fluctuations in the sensory evidence which we estimated through motion energy filtering (*Adelson and Bergen, 1985*; *Urai and Wimmer, 2016*; *Figure 7—figure supplement 1*). These fluctuating motion energy estimates were used as time-varying sensory input to the models, providing key additional constraints over and above nominal sensory evidence levels, choices and RT distributions (*Brunton et al., 2013*).

We first re-fit the standard DDM where the two biasing parameters were allowed to vary with previous choice (see *Figure 1*), now using single-trial motion energy estimates and a non-hierarchical fitting procedure (see Materials and methods). This made these fits directly comparable to both the hierarchical fits in *Figures 3–4*, and the more complex models described below. As expected (*Figure 3a*), the data were better explained by a history-dependent bias in the drift, rather than the starting point (*Figure 7b1*). In these non-hierarchical fits, the hybrid DDM (i.e. both bias terms free

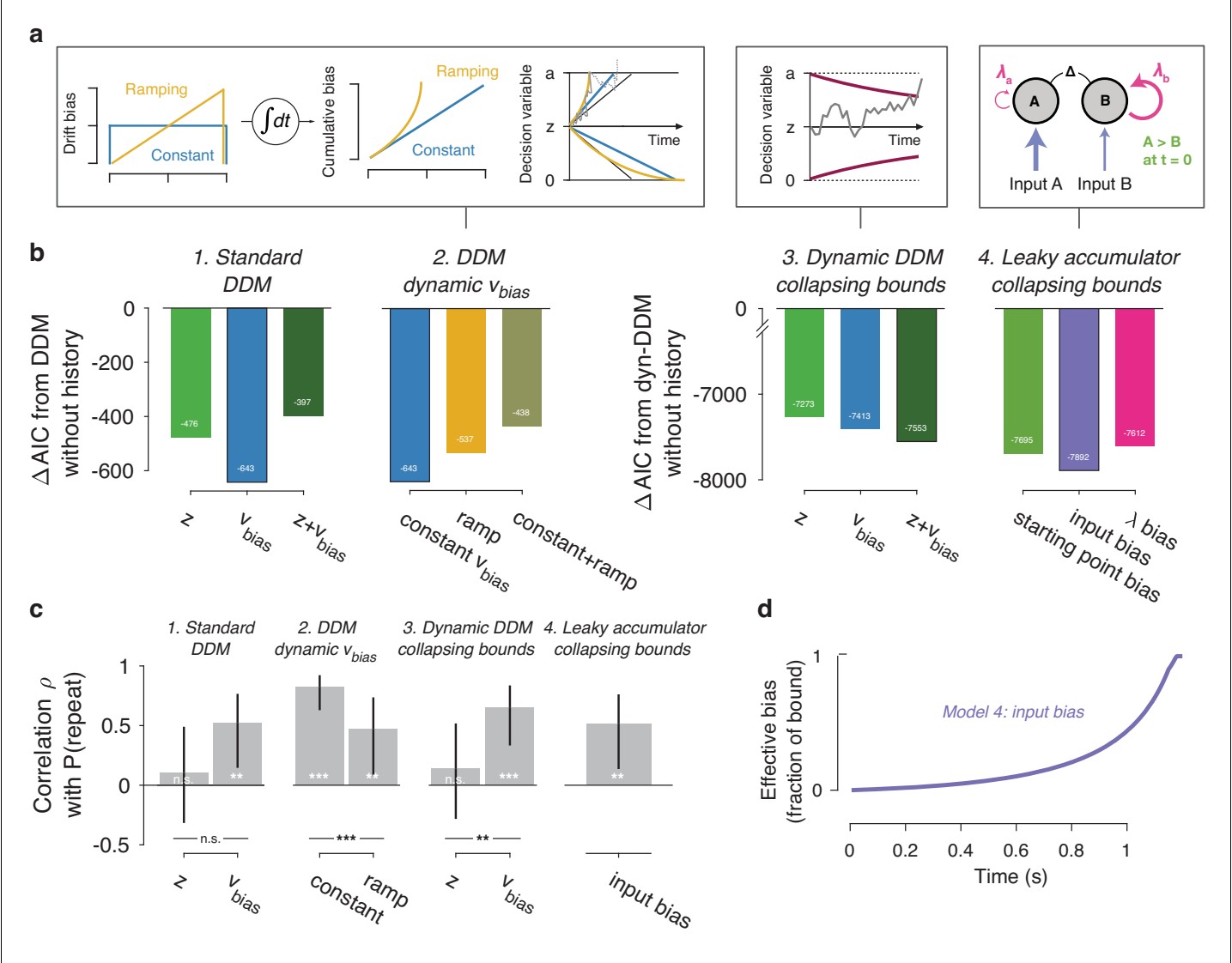

**Figure 7.** Extended dynamic models of biased evidence accumulation. (**a**) Model schematics. In the third panel from the left, the stimulus-dependent mean drift is shown in black, overlaid by the biased mean drift in color (as in *Figure 1a,b*). (**b**) AIC values for each history-dependent model, as compared to a standard (left) or dynamic (right) DDM without history. The winning model (lowest AIC value) within each model class is shown with a black outline. (**c**) Correlation (Spearman's ρ) of parameter estimates with individual repetition behavior, as in *Figure 4b*. Error bars, 95% confidence interval. ***p<0.0001, **p<0.01, n.s. p>0.05. (**d**) Within-trial time courses of effective bias (cumulative bias as a fraction of the decision bound) for the winning leaky accumulator model. Effective bias time courses are indistinguishable between both dynamical regimes (λ < 0 and λ > 0) and are averaged here.

DOI: https://doi.org/10.7554/eLife.46331.016

The following figure supplements are available for figure 7:

**Figure supplement 1.** Motion energy filtering, psychophysical kernels and the effective time-constant of evidence integration.
DOI: https://doi.org/10.7554/eLife.46331.017
**Figure supplement 2.** Leaky accumulator model simulators.
DOI: https://doi.org/10.7554/eLife.46331.019
**Figure supplement 3.** Drift diffusion model simulations.
DOI: https://doi.org/10.7554/eLife.46331.018

to vary as a function of previous choice) lost against the drift bias-only model (indicated by its higher AIC). Yet thi hybrid model allowed for a direct comparison of the correlations between these (jointly fit) bias parameters and individual choice repetition probability. As in our previous analysis (*Figure 4*), individual choice repetition probability was more strongly predicted by drift than starting point bias (*Figure 6c1*).

A previous study of reward effects on speeded decisions reported that reward asymmetries induced supra-linear bias dynamics (*Afacan-Seref et al., 2018*). Temporal integration of a constant drift bias produces a linearly growing effective bias in the decision variable (*Figure 1b*), whereas integration of a ramping drift bias produces a supra-linear growth of effective bias (*Figure 7a*, yellow). In our data, a standard DDM with constant drift bias provided a better fit than DDMs with either a ramping drift bias, or a combination of constant and ramping drift bias (*Figure 7b2*). Furthermore, in the latter (hybrid) model, the constant drift bias best predicted individual choice repetition behavior (*Figure 7c2*), in line with the constant accumulation bias inferred from the standard DDM fits. For the fits shown in *Figure 7b2/c2*, we used the same fitting protocol as for the standard DDM, in which the time-varying sensory evidence fluctuations during stimulus presentation were replaced by their average over time to compute a single-trial drift rate (called 'default protocol', Materials and methods section Extended bounded accumulation models: General assumptions and procedures). The same qualitative pattern of results also held for another fitting protocol ('dynamic protocol', see Materials and methods), in which the time-varying sensory evidence was fed into the integrator (ΔAIC relative to no-history model: −1103, –985, −995, for constant drift bias, ramping drift bias, and hybrid, respectively; correlation with P(repeat): ρ(30)= 0.5458, p=0.0012; ρ(30)= 0.3600, p=0.0429 for constant and ramping drift bias, respectively). We next used this dynamic protocol for a set of more complex dynamical models.

It has been proposed that decision bounds might collapse over time, implementing an 'urgency signal' (*Figure 6a*, middle; *Churchland et al., 2008*; *Cisek et al., 2009*). Indeed, adding collapsing bounds substantially improved our model fits (*Figure 7b3*). This indicates the presence of a strong urgency signal in this task, which had a relatively short stimulus presentation (750 ms) and a tight response deadline (1.25 s after stimulus offset). Critically, a history-dependent drift bias best fit the data (*Figure 7b3*) and captured individual choice repetition behavior (*Figure 7c3*) also in the DDM with collapsing bounds. In other words, while there is evidence for collapsing bounds in this dataset, our conclusion about the impact of history bias on decision formation does not depend on its inclusion in the model.

In the brain, a neural representation of the momentary sensory evidence feeds into a set of accumulators. These consist of circuits of excitatory and inhibitory populations of cortical neurons, which give rise to persistent activity and competitive winner-take-all dynamics (*Usher and McClelland, 2001*; *Wang, 2002*). Under certain parameter regimes, these circuit dynamics can be reduced to lower-dimensional models (*Bogacz et al., 2006*; *Wong, 2006*). In such models, the effective accumulation time constant $1/\lambda$ (with $\lambda$ being the effective leak) results from the balance of leak within each accumulator (due to self-excitation and passive decay) and mutual inhibition between two accumulators encoding different choices (*Usher and McClelland, 2001*). Evidence accumulation can then be biased through an internal representation of the sensory input, or through the way this sensory representation is accumulated (*Figure 7a*, right). We here used a reduced competing accumulator model, where the decision variable was computed as the difference of two leaky accumulators (*Busemeyer and Townsend, 1993*; *Zhang and Bogacz, 2010*; see also *Brunton et al., 2013*) to compare these two accumulation biases and a biased accumulator starting point.

We fit a family of bounded, leaky accumulator models, in which the starting point of the accumulators, their input, or their effective leak $\lambda$ could be biased as a function of previous choice (*Figure 7a*, right). Note that a bias of the accumulator starting point would also translate into an accumulation bias, due to the model dynamics (see Materials and methods section Extended bounded accumulation models: General assumptions and procedures). Even so, comparing this regime with other two biasing mechanism was informative. Also note that we here use the term 'leaky accumulator model' to denote that the model dynamics consisted of a free effective leak parameter $\lambda$, without implying that $\lambda < 0$ (corresponding to activation decay). Our fits allowed $\lambda$ to take either negative ('forgetful' regime) or positive ('unstable' regime) values (*Figure 7—figure supplement 1d*; see also *Brunton et al., 2013*). Critically, in order to test for choice history-dependent accumulation bias, we allowed $\lambda$ of each accumulator to vary as a function of the previous choice,

before computing the difference between the two accumulator activations. Choice-history dependent biases in accumulator starting point or accumulator input were directly applied to the accumulator difference (akin to starting point and drift bias within the DDM). Due to the simplicity of its dynamics, the DDM cannot distinguish between input and leak bias. Indeed, when simulating behavior of leaky accumulator models with either of these two accumulation biases and fitting it with the DDM, both input and λ bias loaded onto DDM drift bias (*Figure 7—figure supplement 2*). Critically, the leaky accumulator with biased accumulator *input* best explained the data, among all the models considered (*Figure 7b4*). Furthermore, the individually estimated input bias predicted individual choice repetition (*Figure 7c4*). This suggests that choice history might specifically bias the internal representation of sensory evidence feeding into the evidence accumulation process.

## Dynamics of effective bias signal approximates rational combination of prior information with current evidence

Taken together, fits and simulations of more complex models provided additional insight into the mechanism underlying choice history bias. They also corroborated the conclusion that choice history biases are mediated by a biased accumulation of evidence, rather than a biased starting point. As a final step, we estimated the time course of the effective bias, computed as the fraction of cumulative bias signal and bound height (*Hanks et al., 2011*). We simulated this signal based on the group average parameters for the best-fitting leaky accumulator model (*Figure 7d*). In this leaky accumulator (with collapsing bound), the effective bias accelerated (*Figure 7d*).

The reader may notice that these (supra-linear) effective bias dynamics are similar to those predicted by the DDM with a ramping drift bias (*Figure 7a*, left). Thus, the observation that the latter model lost by a wide margin against the two models with more complex dynamics (*Figure 7b*, see also Materials and methods) is likely due to features of the data other than the (relatively small) selective history bias. Specifically, the RT distributions were strongly shaped by the urgency signal incorporated by the bound collapse. In the overall best-fitting model (leaky accumulator with collapsing bounds and input bias, *Figure 7b5*), this effective bias depends on the combined effect of two non-linear signals: (i) the cumulative bias resulting from the accumulation of biased input and (ii) the hyperbolically collapsing bound. In the current fits, the effective bias was dominated by the strong bound collapse, but in different circumstances (with weaker urgency signal and for λ < 0), a biased input leaky accumulator can produce a decelerating effective bias. Combination of a biased input with some starting point and or leak bias can further change the dynamics. The key observation is that, regardless of the modeling framework used, we identified an effective bias signal that grew steadily throughout decision formation, in line with the main conclusion drawn from the basic fits of the standard DDM.

Our results are in line with the idea the impact of choice history bias on decision formation grows as a function of elapsed time. This observation might be surprising, as prior information (here: about the previous choice) does not change over time. Yet, previous work has identified a principled rationale for such a time-dependent combination of prior and evidence. When evidence reliability changes from trial to trial, prior information (bias) should be weighted more strongly when sensory evidence is unreliable (*Hanks et al., 2011*; *Moran, 2015*). This can be achieved by increasing the weight of the prior throughout the trial, using elapsed time as a proxy for evidence reliability. This prediction was confirmed experimentally for explicit manipulations of prior probability of the choice options (*Hanks et al., 2011*). Indeed, within the framework of the DDM, this way of combining prior information with current evidence maximizes reward rate (*Moran, 2015*; see also *Drugowitsch and Pouget, 2018*). Only when evidence reliability is constant across trials should prior information be incorporated as a static bias (i.e. starting point). Evidence reliability likely varied from trial to trial across all our experiments (*Moran, 2015*), due to variations in the external input (i.e. mean drift rate in the DDM), originating from stochastically generated stimuli, or internal factors (i.e. drift rate variability in the DDM), such as the inherent variability of sensory cortical responses (*Arieli et al., 1996*; *Faisal et al., 2008*). In particular, the dataset from *Figure 7* entailed strong trial-to-trial variations in the external input (*Figure 7—figure supplement 1*). Thus, the dynamics of the effective bias signal uncovered in *Figure 7d* suggest that participants combined prior information with current evidence in a rational fashion.

## Discussion

Quantitative treatments of perceptual decision-making commonly attribute trial-to-trial variability of overt choices to noisy decision computations (*Shadlen et al., 1996*; *Renart and Machens, 2014*; *Wyart and Koechlin, 2016*). Those accounts typically assume that systematic decision biases remain constant over time. Instead, the choice history biases studied here vary continuously over the course of the experiment, as a function of the previous choices (and choice outcome information). Our current results indicate that choice history explains trial-to-trial variability specifically in evidence accumulation, in a number of widely used perceptual choice tasks. Ignoring such trial-to-trial variations will lead to an overestimation of the noise in the evidence accumulation process and resulting behavior.

History biases in perceptual choice have long been known in perceptual psychophysics (*Fernberger, 1920*) and neuroscience (*Gold et al., 2008*). However, the underlying dynamic mechanisms have remained elusive. We here show that individual differences in overt choice repetition behavior are explained by the degree to which choices bias the evidence accumulation, not the starting point, of subsequent decisions. This accumulation bias is associated with choices made several trials into the past, and it grows steadily as the current decision unfolds. This insight calls for a revision of current models of choice history biases (*Yu and Cohen, 2008*; *Zhang et al., 2014*).

It is instructive to relate our results to previous studies manipulating the probability of the occurrence of a particular category (i.e. independently of the *sequence* of categories) or the asymmetry between rewards for both choices. Most of these studies explained the resulting behavioral biases in terms of starting point shifts (*Leite and Ratcliff, 2011*; *Mulder et al., 2012*; *White and Poldrack, 2014*; *Rorie et al., 2010*; *Gao et al., 2011*; but only for decisions without time pressure, see *Afacan-Seref et al., 2018*). Yet, one study with variations of evidence strength found an effect of asymmetric target probability on accumulation bias (*Hanks et al., 2011*) similar to the one we here identified for choice history. In all this previous work, biases were under experimental control: probability or reward manipulations were signaled via explicit task instructions or single-trial cues (in humans) or block structure (in animals). By contrast, the choice history biases we studied here emerge spontaneously and in an idiosyncratic fashion (*Figure 2e*), necessitating our focus on individual differences.

Our modeling addressed the question of how prior information is combined with new evidence during decision formation (see in particular the section Dynamics of effective bias signal approximates rational combination of prior information with current evidence). But why did participants use choice history as a prior for their decisions? In all our experiments, the sensory evidence was uncorrelated across trials – as is the case in the majority of perceptual choice tasks used in the literature. Thus, any history bias can only reduce performance below the level that could be achieved, given the observer's sensitivity. It may seem irrational that people use history biases in such settings. However, real-world sensory evidence is typically stable (i.e. auto-correlated) across various timescales (*Yu and Cohen, 2008*). Thus, people might (erroneously) apply an internal model of this environmental stability to randomized laboratory experiments (*Yu and Cohen, 2008*), which will push them toward choice repetition or alternation (*Glaze et al., 2015*). Indeed, people flexibly adjust their choice history biases to environments with different levels of stability (*Glaze et al., 2015*; *Kim et al., 2017*; *Braun et al., 2018*), revealing the importance of such internal models on perceptual decision-making. In sum, with our conclusions from the time course of the effective bias signal, these considerations suggest that participants may have applied a rational strategy, but based on erroneous assumptions about the structure of the environment.

While we found that choice history-dependent variations of accumulation bias were generally more predictive of individual choice repetition behavior, the DDM starting point was consistently shifted away from the previous response for a majority of participants (i.e. negative values along x-axis of *Figure 4a*). This shift was statistically significant in three out or six datasets (*Figure 4—figure supplement 5a*), and might explain the advantage of the dual parameter model over the pure drift-bias model in our model comparisons (*Figure 3a*). The starting point shift may be due to at least two scenarios, which are not mutually exclusive. First, it might reflect a stereotypical response alternation tendency originating from neural dynamics in motor cortex – for example, a post-movement 'rebound' of beta-band oscillations (*Pfurtscheller et al., 1996*). Indeed, previous work found that beta rebound is related to response alternation in a perceptual choice task, which precluded (in

contrast to our tasks) motor planning during evidence accumulation (*Pape and Siegel, 2016*). This stereotypical response alternation tendency (via starting point) may have conspired with the more flexible history bias of evidence accumulation (via drift bias) to shape choice behavior. Because starting point shifts will predominantly contribute to fast decisions, this scenario is consistent with the average choice alternation tendency we observed for RTs < 600 ms (*Figure 4—figure supplement 5c*). Because the response alternation tendency in motor cortex is likely to be induced only by the immediately preceding response, this scenario is also consistent with the shorter timescales we estimated for the starting point effects (1.39 trials) than the drift rate effects (2.38 trials; *Figure 6a*, exponential fits). Second, the starting point shift may also reflect decision dynamics more complex than described by the standard DDM: non-linear drift biases (*Figure 7—figure supplement 2*, third column) or biases in the leak of decision accumulators (*Figure 7—figure supplement 3*, third column). Both give rise to opposite effects on drift bias and starting point bias when fit with the standard DDM, thus yielding negative correlations between DDM starting point and drift bias estimates. Such negative correlations were present in our data, but weak and not statistically significant (Spearman's rho −0.4130 to −0.0757, combined $BF_{10}$= 0.0473). It is possible that both of the scenarios discussed here conspired to yield the starting point effects observed in model comparisons and individual parameter estimates. Future work is needed to illuminate this issue, for example through manipulating decision speed and/or the delays between subsequent motor responses, and modeling choice-related neural dynamics in motor cortex.

We propose that choice history biases evidence accumulation, but there are alternative scenarios. First, it is possible that participants' choices were due to computations altogether different from those incorporated in the bounded accumulation models assessed here. All our models imply simple neural accumulators with persistent activity. At least on a subset of trials, participants may make fast guesses (*Noorbaloochi et al., 2015*), or engage in automatic decision processing (*Servant et al., 2014*; *Ulrich et al., 2015*) or post-accumulation biases (*Erlich et al., 2015*). The decision computation may also entail noise-driven attractor dynamics (*Wang, 2002*; *Braun and Mattia, 2010*) possibly with sudden 'jumps' between neural activity states (*Latimer et al., 2015*), instead of linear accumulation to a threshold level. Even if the accumulation dynamics postulated in our models cannot be reduced to the dynamics of single neurons, the history-dependent accumulation bias we inferred here would constitute a valid description of the collective computational properties of the neural system producing choice behavior. Second, within bounded accumulation models, any directed change in the decision variable can be mimicked by some selective (i.e. asymmetric) change in one of the decision bounds. For example, combining the DDM with a linearly collapsing bound for the favored choice and a linearly expanding bound for the other choice has the same effect on choice fractions and RT distributions as a drift bias. We are not aware of any empirical evidence for such asymmetric changes in decision bounds. Decision-related cortical ramping activity seems to always reach a fixed level just before motor response, irrespective of prior probabilities (*Hanks et al., 2011*) or speed-accuracy trade-offs (*Hanks et al., 2014*; *Murphy et al., 2016*). Instead, the build-up of this activity is biased by prior information (*Hanks et al., 2011*).

A plausible mechanism underlying the choice history-dependent shift in accumulation bias is a bias of the neural representations of the sensory evidence towards (or away from) a previously selected category (*Nienborg and Cumming, 2009*; *St John-Saaltink et al., 2016*; *Urai and Wimmer, 2016*). This is precisely the 'input bias' scenario entailed in our best fitting model (*Figure 7*). The primate brain is equipped with powerful machinery to bias sensory representations in a top-down fashion (*Desimone and Duncan, 1995*; *Reynolds and Heeger, 2009*). In the laboratory, these top-down mechanisms have been probed by explicitly instructing subjects to shift their attention to a particular sensory feature or location. Such instructions induce biased activity states in regions of prefrontal and parietal association cortex, which are propagated down the cortical hierarchy to sensory cortex via selective feedback projections, where they boost the corresponding feature representations and suppress others (*Desimone and Duncan, 1995*). The same prefrontal and parietal regions accumulate sensory evidence and seem to carry choice history signals. It is tempting to speculate that choice history signals in these regions cause the same top-down modulation of sensory cortex as during explicit manipulations of attention. In other words, agents' choices might be one factor directing their top-down attention under natural conditions, in a way analogous to explicit attention cues in laboratory tasks. An alternative, but related possibility is that the direction of selective attention fluctuates spontaneously during the course of a perceptual choice experiment,

preferentially sampling features supporting one choice for a streak of trials, and then switching to sampling support for the other category. The corresponding top-down modulations would bias evidence accumulation and choice in a serially correlated fashion. These ideas are not mutually exclusive and can be tested by means of multi-area neurophysiological recordings combined with local perturbations.

A growing body of evidence points to the interplay of multiple timescales for neural computation in the cortex. One line of behavioral work has revealed effective (within-trial) evidence accumulation over timescales ranging from a few hundred milliseconds (*Kiani et al., 2008*; *Tsetsos et al., 2015*) to several seconds (*Tsetsos et al., 2012*; *Wyart et al., 2012*; *Cheadle et al., 2014*). Another line of work, including the current study, revealed the slow accumulation of internal decision variables or external outcome information across trials (tens of seconds) to build up time-varying biases, or priors (*Sugrue et al., 2004*; *Abrahamyan et al., 2016*; *Purcell and Kiani, 2016b*; *Braun et al., 2018*). Relatedly, neurophysiological work on ongoing activity has inferred multiple hierarchically organized timescales in different cortical regions (*Honey et al., 2012*; *Murray et al., 2014*; *Chaudhuri et al., 2015*; *Runyan et al., 2017*; *Scott et al., 2017*). The history-dependent evidence accumulation biases that we have uncovered here might index the interplay between these different effective timescales, with long-timescale accumulators at higher stages biasing short-timescale accumulators at intermediate stages of the cortical hierarchy.

## Materials and methods

### Datasets: behavioral tasks and participants

We analyzed six different datasets, four of which were previously published. These spanned different modalities (visual or auditory), decision-relevant sensory features (motion direction, contrast, tone presence, motion coherence), and tasks (detection or discrimination). In each dataset, the number of participants was determined to allow for robust estimation of the original effects of interest. No participants were excluded from the analyses.

Those tasks where the decision-relevant sensory evidence was presented until the observer generated a response were called response time (RT) tasks; those tasks where the sensory evidence was presented for a fixed duration, and its offset cues the observer's response, were called fixed duration (FD) tasks in line with the terminology from *Mazurek et al. (2003)*. These two protocols have also been termed 'free response protocol' and 'interrogation protocol' (*Bogacz et al., 2006*). In all datasets, stimulus strength (i.e., decision difficulty) was kept constant, or varied systematically across levels, within all main experimental sessions that were used for fitting the DDM.

### 2AFC visual motion direction discrimination task (RT)

These data were previously published (*Murphy et al., 2014*), and are available at https://doi.org/10.5061/dryad.tb542. The study was approved by the ethics committee of the Leiden University Cognitive Psychology department, and all subjects provided written informed consent before taking part. Twenty-six observers (22 women and 4 men, aged 18–29) performed a motion direction (left vs. right) discrimination task. Stationary white dots were presented on a black screen for an interval of 4.3–5.8 s. After this fixation interval, the decision-relevant sensory evidence was presented: some percentage of dots (the 'motion coherence' level) moved to the left or the right. The coherence was individually titrated to yield an accuracy level of 85% correct (estimated from a psychometric function fit) before the start of the main experiment, and kept constant afterwards. The moving dots were presented until observers indicated their choice with a button press. After the response, the fixation cross changed color for 700 ms to indicate single-trial feedback. Each observer performed 500 trials of the task in one session. We refer to this task as 'Visual motion 2AFC (RT)'.

### 2AFC visual motion direction discrimination task (FD)
Participants and informed consent
Thirty-two participants (aged 19–35 years, 43 women and 21 men) participated in the study after giving their informed consent. The experiment was approved by the ethical review board of the University Medical Center Hamburg-Eppendorf (PV4714).

## Task and procedure

Observers performed a fixed duration version of the random dot motion discrimination (up vs. down) task in the MEG scanner. White dots were displayed on a gray background screen, with a density of 6 dots/degree$^2$, resulting in 118 dots on the screen at each frame. The stimuli were confined to a circle of 2.5° radius, which was placed in the lower half of the visual field at 3.5° from fixation. After a fixation interval of 0.75–1.5 s, random dot motion stimuli (0, 3, 9, 27 or 81% motion coherence) were displayed for 750 ms. Signal dots moved with a speeds of 11.5 degree/s, and noise dots were randomly displaced within the circle on each frame. We used the single-trial dot coordinates to construct time courses of fluctuating external evidence (see Materials and methods section *Motion energy filtering and psychophysical kernels*; *Figure 7—figure supplement 1a–c*). Observers received auditory feedback 1.5–2.5 s after their response, and the ISI started 2–2.5 s after feedback. Observed performed 1782 trials over three sessions, in which the stimulus transition probability varied (0.2, 0.5 or 0.8) between blocks of 99 trials. To maximize trial counts for the non-hierarchical leaky accumulator fits, we here collapsed across blocks. We refer to this task as 'Visual motion 2AFC (FD)'.

## Visual motion coherence discrimination 2IFC task (FD): dataset 1

These data were previously published in *Urai et al. (2017)*, and are available at http://dx.doi.org/10.6084/m9.figshare.4300043. The ethics committee at the University of Amsterdam approved the study, and all observers gave their informed consent before participation. Twenty-seven observers (17 women and 10 men, aged 18–43) performed a two-interval motion coherence discrimination task. They viewed two consecutive intervals of random dot motion, containing coherent motion signals in a constant direction towards one of the four diagonals (counterbalanced across participants) and judged whether the second test interval (variable coherence) contained stronger or weaker motion than the first reference (constant coherence) interval. After a fixation interval of 0.5–1 s, they viewed two consecutive intervals of 500 ms each, separated by a delay of 300–700 ms. The decision-relevant sensory evidence (i.e. the difference in motion coherence between intervals), was chosen pseudo-randomly for each trial from the set (0.625, 1.25, 2.5, 5, 10, 20, 30%). Observers received auditory feedback on their choice after a delay of 1.5–2.5 s. After continuing to view noise dots for 2–2.5 s, stationary dots indicated an inter-trial interval. Observers self-initiated the start of the next trial (range of median inter-trial intervals across observers: 0.68–2.05 s). Each observer performed 2500 trials of the task, divided over five sessions. We refer to this task as 'Visual motion 2IFC (FD) #1'.

## 2IFC visual motion coherence discrimination task (FD): dataset 2

### Participants and informed consent

Sixty-two participants (aged 19–35 years, 43 women and 19 men) participated in the study after screening for psychiatric, neurological or medical conditions. All subjects had normal or corrected to normal vision, were non-smokers, and gave their informed consent before the start of the study. The experiment was approved by the ethical review board of the University Medical Center Hamburg-Eppendorf (PV4648).

### Task protocol

Observers performed five sessions, of which the first and the last took place in the MEG scanner (600 trials divided over 10 blocks per session) and the three sessions in between took place in a behavioral lab (1500 trials divided over 15 blocks per session). The task was as described above for 'Visual motion 2IFC (FD) #1', with the following exceptions. The strength of the decision-relevant sensory evidence was individually titrated to an accuracy level of 70% correct, estimated from a psychometric function fit, before the start of the main experiment and kept constant for each individual throughout the main experiment. Each stimulus was presented for 750 ms. In the MEG sessions, auditory feedback was presented 1.5–3 s after response, and an inter-trial interval with stationary dots started 2–3 s after feedback. Participants initiated the next trial with a button press (across-subject range of median inter-trial interval duration: 0.64 to 2.52 s, group average: 1.18 s). In the training sessions, auditory feedback was presented immediately after the response. This was followed by an inter-trial interval of 1 s, after which the next trial started. In this experiment, three sub-groups of

observers received different pharmacological treatments prior to each session, receiving placebo, atomoxetine (a noradrenaline reuptake inhibitor), or donepezil (an acetylcholinesterase inhibitor). These groups did not differ in their choice history bias and were pooled for the purpose of the present study (*Figure 4—figure supplement 3*). We refer to this task as 'Visual motion 2IFC (FD) #2'.

## Visual contrast yes/no detection task (RT)

These data were previously published (*de Gee et al., 2014*), and are available at https://doi.org/10.6084/m9.figshare.4806559. The ethics committee of the Psychology Department of the University of Amsterdam approved the study. All participants took part after giving their written informed consent. Twenty-nine observers (14 women and 15 men, aged 18–38) performed a yes/no contrast detection task. During a fixation interval of 4–6 s, observers viewed dynamic noise (a binary noise pattern that was refreshed each frame, at 100 Hz). A beep indicated the start of the decision-relevant sensory evidence. On half the trials, a vertical grating was superimposed onto the dynamic noise; on the other half of trials, only the dynamic noise was shown. The sensory evidence (signal +noise or noise-only) was presented until the observers reported their choice ('yes', grating was present; or 'no', grating was absent), or after a maximum of 2.5 s. The signal contrast was individually titrated to yield an accuracy level of 75% correct using a method of constant stimuli before the main experiment, and kept constant throughout the main experiment. Observers performed between 480–800 trials over 6–10 sessions. Six observers in the original paper (*de Gee et al., 2014*) performed a longer version of the task in which they also reported their confidence levels and received feedback; these were left out of the current analysis, leaving 23 subjects to be included. We refer to this task as 'Visual contrast yes/no (RT)'.

## Auditory tone yes/no detection task (RT)

These data were previously published (*de Gee et al., 2017*) and are available at https://doi.org/10.6084/m9.figshare.4806562. All subjects gave written informed consent. The ethics committee of the Psychology Department of the University of Amsterdam approved the experiment. Twenty-four observers (20 women and four men, aged 19–23) performed an auditory tone detection task. After an inter-trial interval of 3–4 s, decision-relevant sensory evidence was presented: on half the trials, a sine wave (2 KHz) superimposed onto dynamic noise (so-called TORCS; *McGinley et al., 2015*) was presented; on the other half of trials only the dynamic noise was presented. The sensory evidence was presented until the participant reported their choice button press or after a maximum of 2.5 s. No feedback was provided. Each individual's signal volume was titrated to an accuracy level of 75% correct using an adaptive staircase procedure before the start of the main experiment, and kept constant throughout the main experiment. Participants performed between 1320 and 1560 trials each, divided over two sessions. We refer to this task as 'Auditory yes/no (RT)'.

## Model-free analysis of sensitivity and choice history bias

We quantified perceptual sensitivity in terms of signal detection-theoretic d' (*Green and Swets, 1966*):

$$d' = \Phi^{-1}(H) - \Phi^{-1}(FA) \tag{1}$$

where $\Phi$ was the normal cumulative distribution function, H was the fraction of hits and FA the fraction of false alarms. In the 2AFC and 2IFC datasets, one of the two stimulus categories was arbitrarily treated as signal absent. Both H and FA were bounded between 0.001 and 0.999 to allow for computation of d' in case of near-perfect performance (*Stanislaw and Todorov, 1999*). We estimated d' separately for each individual and, for the two datasets with varying difficulty levels, for each level of sensory evidence.

We quantified individual choice history bias in terms of the probability of repeating a choice, termed P(repeat), regardless of the category of the (previous or current) stimulus. This yielded a measure of bias that ranged between 0 (maximum alternation bias) and 1 (maximum repetition bias), whereby 0.5 indicated no bias.

## Drift diffusion model (DDM) fits

### General

This section describes the general DDM, with a focus on the biasing mechanisms described in Results and illustrated in *Figure 1* (*Ratcliff and McKoon, 2008*). Ignoring non-decision time, drift rate variability, and starting point variability (see below), the DDM describes the accumulation of noisy sensory evidence:

$$dy = s \cdot v \cdot dt + cdW \qquad (2)$$

where $y$ is the decision variable (gray example traces in *Figure 1*), $s$ is the stimulus category (coded as -1,1), v is the drift rate, and $cdW$ is Gaussian distributed white noise with mean 0 and variance $c^2 dt$ (*Bogacz et al., 2006*). In an unbiased case, the starting point of the decision variably $y(0) = z$, is situated midway between the two decision bounds 0 and $a$:

$$y(0) = z = \frac{a}{2} \qquad (3)$$

where $a$ is the separation between the two decision bounds. A bias in the starting point is implemented by an additive offset $z_{bias}$ from the midpoint between the two bounds (*Figure 1a*):

$$y(0) = z = \frac{a}{2} + z_{bias} \qquad (4)$$

A drift bias can be implemented by adding a stimulus-independent constant $v_{bias}$, also referred to as drift bias (*Ratcliff and McKoon, 2008*), to the (stimulus-dependent) mean drift (*Figure 1b*). This adds a bias to the drift that linearly grows with time:

$$dy = (s \cdot v + v_{bias})dt + cdW \qquad (5)$$

We allowed both bias parameters to vary as a function of observers' previous choice. These two biasing mechanisms result in the same (asymmetric) fraction of choices, but they differ in terms of the resulting shapes of RT distributions (*Figure 1*). In previous work, $z_{bias}$ and $v_{bias}$ have also been referred to as 'prior' and 'dynamic' bias (*Moran, 2015*) or 'judgmental' and 'perceptual' bias (*Liston and Stone, 2008*).

### Estimating HDDM Bias parameters

We used hierarchical drift diffusion modeling as implemented in the HDDM toolbox (*Wiecki et al., 2013*) to fit the model and estimate its parameters. As recommended by the HDDM toolbox, we specified 5% of responses to be contaminants, meaning they arise from a process other than the accumulation of evidence - for example, a lapse in attention (*Ratcliff and Tuerlinckx, 2002*). We fit the DDM to RT distributions for the two choice categories, conditioned on the stimulus category for each trial ($s$ in *Equation 2*) - a procedure referred to as 'stimulus coding'. This fitting method deviates from a widely used expression of the model, where RT distributions for correct and incorrect choices are fit (also called 'accuracy coding'). Only the former can fit decision biases towards one choice over the other.

First, we estimated a model without history-dependence. Overall drift rate, boundary separation, non-decision time, starting point, and drift bias were estimated for each individual (*Figure 3—figure supplement 1*). Across-trial variability in drift rate and starting point were estimated at the group-level only (*Ratcliff and Childers, 2015*). For the datasets including variations of sensory evidence strength (Visual motion 2AFC (FD) and Visual motion 2IFC (FD) #1), we separately estimated drift rate for each level of evidence strength. This model was used to confirm that the DDM was able to fit all datasets well, and to serve as a baseline for model comparison.

Second, we estimated three different models of history bias, allowing (i) starting point, (ii) drift or (iii) both to vary as a function of the observer's immediately preceding choice (thus capturing only so-called first-order sequential effects; cf *Gao et al., 2009*; *Wilder et al., 2009*). The effect of the preceding choice on each bias parameter was then termed its 'history shift'. For example, for the visual motion direction discrimination task we separately estimated the starting point parameter for trials following 'left' and 'right' choices. The difference between these two parameters then reflected

individual observers' history shift in starting point, computed such that a positive value reflected a tendency towards repetition and a negative value a tendency towards alternation. The history shift in drift bias was computed in the same way.

## HDDM regression models

We estimated the effect of up to six previous stimuli and choices on history bias using a HDDM regression model. We first created a design matrix $X$ with dimensions *trials x 2 * lags*, which included pairs of regressors coding for previous stimuli and choices (coded as $-1, 1$), until (and including) each model's lag. Two distinct replicas of $X$ were then used as design matrices to predict drift bias ($X_v$) and starting point ($X_z$). Drift bias was defined as $v \sim 1 + s + X_v$, where 1 captured an overall bias for one choice over the other and $s$ indicated the signed stimulus strength. Starting point was defined as $z \sim 1 + X_z$, with a logistic link function $\frac{1}{1+e^{-x}}$.

After fitting, parameter estimates were recombined to reflect the effect of previous correct (choice + stimuli) or error (choice – stimuli) trials. We sign-flipped the weight values for alternators (i.e. those participants with a repetition tendency at lag one < 0.5); this makes all the panels in *Figure 6* a interpretable as a change in each parameter in the direction of individual history bias.

## HDDM model fitting procedures

The HDDM (*Wiecki et al., 2013*) uses Markov-chain Monte Carlo sampling for generating posterior distributions over model parameters. Two features of this method deviate from more standard model optimization. First, the Bayesian MCMC generates full posterior distributions over parameter estimates, quantifying not only the most likely parameter value but also the uncertainty associated with that estimate. Second, the hierarchical nature of the model assumes that all observers in a dataset are drawn from a group, with specific group-level prior distributions that are informed by the literature (*Figure 3—figure supplement 1*; *Wiecki et al., 2013*). In practice, this results in more stable parameter estimates for individual subjects, who are constrained by the group-level inference. Note that we also repeated our model fits with more traditional $G^2$ optimization (*Ratcliff and Tuerlinckx, 2002*) and obtained similar results (*Figure 4—figure supplement 2a*).

For each variant of the model, we ran 30 separate Markov chains with 5000 samples each. Of those, half were discarded as burn-in and every second sample was discarded for thinning, reducing autocorrelation in the chains. This left 1250 samples per chain, which were concatenated across chains. Individual parameter estimates were then estimated from the posterior distributions across the resulting 37500 samples. All group-level chains were visually inspected to ensure convergence. Additionally, we computed the Gelman-Rubin $\hat{R}$ statistic (which compares within-chain and between-chain variance) and checked that all group-level parameters had an $\hat{R}$ between 0.9997 and 1.0406.

Formal comparison between the different model variants was performed using the Akaike Information Criterion (*Akaike, 1974*): $AIC = -2L + 2k$, where $L$ is the total loglikelihood of the model and $k$ denotes the number of free parameters. The AIC was computed for each observer, and summed across them. Lower AIC values indicate a better fit, while taking into account the complexity of each model. A difference in AIC values of more than 10 is considered evidence for the winning model to capture the data significantly better. The conclusions drawn from AIC also hold when using the Deviance Information Criterion for the hierarchical models.

## Conditional bias functions

For each variant of the model and each dataset, we simulated data using the best-fitting parameters. Specifically, we simulated 100 responses (choices and RTs) for each trial performed by the observers. These predicted patterns for the 'baseline model' (without history-dependence) were first used to compare the observed and predicted patterns of choices and RTs (*Figure 3—figure supplement 2*).

We used these simulated data, as well as the participants' choices and RTs, to visualize specific features in our data that distinguish the different biased models (*Palminteri et al., 2017*). Specifically, we computed conditional bias functions (*White and Poldrack, 2014*) that visualize choice history bias as a function of RTs. Each choice was recoded into a repetition (1) or alternation (0) of the previous choice. We then expressed each choice as being either in line with, or against the observer's individual bias (classified into 'repeaters' and 'alternators' depending on choice repetition probability). Note that given the transformation of the data (sign-flipping the bias data for alternators in

order to merge the two groups), the fact the *average* P(bias)>0.5 is trivial, and would occur for any generative model of history bias. Conditional bias functions instead focus on the effect of choice history bias as a function of time within each trial, the shape of which distinguishes between different bias sources (*Figure 1c*).

To generate these conditional bias functions, we divided each (simulated or real) observer's RT distribution into five quantiles (0.1, 0.3, 0.5, 0.7 and 0.9) and computed the fraction of biased choices within each quantile. The shape of the conditional bias functions for models with $z$ and $v_{bias}$ confirm that $z$ predominantly produces biased choices with short RTs, whereas $v_{bias}$ leads to biased choices across the entire range of RTs (*Figure 3b*).

## Motion energy filtering and psychophysical kernels

For the Visual motion 2AFC (FD) dataset, we used motion energy filtering (using the filters described in *Urai and Wimmer, 2016*) to reconstruct the time-course of fluctuating sensory evidence over the course of each individual trial, averaging over the spatial dimensions of the display (*Figure 7—figure supplement 1a, b*). These single-trial traces then served as the time-resolved input to a set of extended DDM and leaky accumulator models (*Figure 7*). Specifically, filtering the stimuli at 60 Hz (the refresh rate of the LCD projector) resulted in 45 discrete samples for the 750 ms viewing period of each trial. The first 13 samples of the motion energy filter output (first 200 ms of the viewing interval) corresponded to the 'rise time' of the filter (*Kiani et al., 2008*), yielding outputs that were a poor representation of the actual motion energy levels (see also *Figure 7—figure supplement 1a*). In order to prevent those uninterpretable filter outputs from contributing, we discarded the first 15 samples (250 ms) before model fitting (see below). Using constant interpolation, we expanded the remaining 30 samples onto 150 samples, which, given that the simulation Euler step was 5 ms (dt= 0.005), corresponded to a 750 ms long input time series. In the model descriptions below we denote the input time series with $M = \{M_t : t \in T\}$ and $T = \{1, 2, \ldots, 150\}$.

We also used these motion energy traces to construct so-called psychophysical kernels. Within each stimulus identity (motion direction and coherence, excluding the easiest 81% coherence trials), we subtracted the average motion energy traces corresponding to 'up' vs. 'down' choices. The resulting trace represents the excess motion energy that drives choices, over and above the generative stimulus coherence (*Figure 7—figure supplement 1c*).

## Extended bounded accumulation models
### General assumptions and procedures

In the 2AFC (FD) visual motion experiment participants viewed the stimulus for 0.75 s (hereafter called 'viewing period') and could respond only after the stimulus offset. This required specifying the input to the evidence accumulation process. In the models described below, we used separate simulation protocols, based on different assumptions about this input. In the 'dynamic' protocol, where the input was the time-varying sensory evidence from each trial, the accumulation process was assumed to start at stimulus onset, and responses could happen during the motion viewing interval. The average activity of the accumulator(s) at stimulus offset served as input for accumulation during the post-offset period. For fitting models using this protocol, empirical RTs were calculated relative to the stimulus onset. Motion energy estimates were used as time-resolved input to the model.

By contrast, in the 'default' protocol, the motion energy fluctuations were averaged across the viewing interval excluding the filter rise time (i.e. from 250 to 750 s after stimulus offset), and the average motion energy was then used as a single-trial drift rate for the accumulation process. In other words, the accumulation-to-bound dynamics only took place during the post-offset period. Accordingly, when fitting models with this protocol, the empirical RTs were calculated relative to stimulus offset. Using this protocol was necessary for replicating our basic result from the standard DDM fits: For the 'dynamic' protocol, any starting point bias would turn into a drift bias because it would feed into accumulation process after stimulus offset, precluding the comparison between the two forms of bias. Thus, we used only the default protocol for the standard DDM fits, which aimed at differentiating between starting point and accumulation biases. For comparison, we also used the same simulation protocol when fitting an extended DDM with a both a constant and a ramping component in the drift bias (see below). We then switched to the more realistic dynamic protocol for the subsequent models with more complex dynamics.

The AIC scores of models using the default protocol were generally lower (i.e. better) compared to the respective models that used the dynamic protocol. This difference is likely due to the fact that the dynamic protocol is more constrained by using as input to the models the exact motion energy traces rather than just their mean for each trial. AIC is blind to such latent flexibility differences that do not map onto differences in number of parameters. Thus, AIC may have 'under-penalized' models in the default protocol relative to those in the dynamic protocol.

In all models and in both simulation protocols, model predictions were derived via Monte Carlo simulation. The variance of the processing noise was set to $c^2 = 1$. One simulation time-step corresponded to 5 ms (Euler step, $dt = 0.005$). Finally, in the standard protocol the accumulation process could last for a maximum of 300 time-steps (or 1500 ms) and in the dynamic protocol for a maximum of 450 time-steps (or 2250 ms). After these time points, the process timed-out and a response was assigned to the alternative according to the state of the diffusion variable (e.g. in the standard DDM right if $y > \frac{a}{2}$ and left if $y < \frac{a}{2}$).

## DDM variants with default simulation protocol

For all basic DDM variants described in this section, we used the default simulation protocol: the time-averaged motion energy for each trial provided the drift-rate ($v$) driving the subsequent diffusion process. DDM models had five generic parameters: threshold ($a$), noise scaling ($g$), non-decision time ($Ter$), drift-rate variability ($sv$) and starting-point variability ($sz$).

*Naïve DDM.* We denote with $y$ the state of the diffusion variable. At time 0:

$$y(0) = z = \frac{a}{2} + U(-sz, sz) \tag{6}$$

where $U$ was a uniform random variable (rectangular distribution) in the $(-sz, sz)$ range. The evolution of $y$ was described by:

$$dy = g \cdot \ddot{v} \cdot dt + cdW \tag{7}$$

Above, $g$ was the scaling parameter that controls the signal-to-noise-ration (given that $c$ is fixed at 1). The variable $\ddot{v}$ was the effective drift-rate, that is a Gaussian variable with $N(m, sz^2)$ where $sz$ was the drift-rate variability and $m$ was the average of the motion energy on each trial. A response was generated when the decision variable $y$ exceeded $a$ (right choice) or surpassed 0 (left choice). The moment that either of these boundaries was crossed plus a non-decision time $Ter$, determined the per-trial RT.

*Starting point DDM.* This model was the same as the naïve model but with an extra parameter $z_{bias}$ such that at time 0:

$$y(0) = \frac{a}{2} + U(-sz, sz) + z_{bias} \cdot prev \tag{8}$$

The variable *prev* here encoded the previous choice (1: right, -1: left). If $z_{bias}$ was positive the model implemented repetition and if negative it implemented alternation.

*Drift bias DDM.* Same as the naïve model but with an extra biasing parameter $v_{bias}$ such that:

$$dy = (g \cdot \ddot{v} + v_{bias} \cdot prev)dt + cdW \tag{9}$$

*Hybrid DDM.* This version combined the starting point DDM and drift bias DDM using two biasing parameters.

*Simple Ramping DDM.* This model was the same as the naïve model but with an extra parameter $s_{ramp}$ such that:

$$dy = \left( g \cdot \ddot{v} + \frac{s_{ramp} \cdot t \cdot prev}{t_{max}} \right) dt + cdW \tag{10}$$

where $t$ denoted time elapsed in terms of Monte-Carlo time-steps and $t_{max}$ = 300 time-steps, which was the maximum duration that a given trial could run for.

*Hybrid Ramping DDM.* Same as the naïve model but with 2 extra parameters $s_{ramp}$ and $s_{constant}$ such that:

$$\mathrm{dy} = \left( \mathrm{g} \cdot \ddot{\mathrm{v}} + \left( s_{constant} + \frac{s_{ramp} \cdot t}{t_{max}} \right) prev \right) \cdot \mathrm{dt} + \mathrm{cdW} \tag{11}$$

This model thus implemented a drift bias that is nonzero at the start of the trial ($s_{constant}$), and also linearly increases until the end of the trial (with slope $s_{ramp}$).

## Extended models with dynamic simulation protocol

For all subsequently described models, we used the dynamic simulation protocol (see section *General Assumptions and Procedures*), with the motion energy time courses serving as input to the accumulation process. To illustrate the details of the dynamic protocol, we next describe how the decision variable was updated in the case of the naïve DDM. The decision variable during the viewing period evolved according to the following differential equation:

$$\mathrm{dy(t)} = \mathrm{g} \cdot M_t \cdot \mathrm{dt} + \mathrm{cdW} \tag{12}$$

where $M_t$ was the value of the input signal at time *t*. Following stimulus offset (at *t* = *T*), after 150 time-steps, the diffusion variable carried on being updated as follows:

$$\mathrm{dy(t)} = \frac{y(T)}{T} + \mathrm{cdW} \tag{13}$$

In other words, after the stimulus disappeared, accumulation was driven by the average evidence accumulated up to the point of stimulus offset. This post-stimulus accumulation could continue for a maximum of 300 extra time-steps, at which point the process timed-out.

*Simple and Hybrid Ramping DDM.* This model was the same as the above Simple and Hybrid Ramping DDMs, only now fit by using the dynamic simulation protocol (i.e. the ramping drift-criterion bias is applied for the viewing period only and, following stimulus offset, the decision variable is updated according to *Equation 13*).

## Dynamic DDM with collapsing bounds

In the 'collapsing bounds' DDM models, a response was generated when the diffusion variable ($y$) exceeds $\mathrm{b}_{up}$ (right choice) or surpasses $\mathrm{b}_{down}$ (left choice). The two thresholds, $\mathrm{b}_{up}$ and $\mathrm{b}_{down}$, vary over time as follows:

$$\mathrm{b}_{up}(t) = \left| a - a\frac{t}{t+c} \right|_{a/2}^{a} \tag{14.1}$$

$$\mathrm{b}_{down}(t) = \left| a\frac{t}{t+c} \right|_{0}^{a/2} \tag{14.2}$$

In the above, the notation $|x|_{min}^{max}$ indicates that *x* was clamped such that $x \in [min, max]$.

The moment that either of these boundaries was reached, plus a non-decision time *Ter*, determined the per-trial RT. The dynamic DDM model had five basic parameters: threshold initial value (*a*), threshold collapse rate (*c*), noise scaling (*g*), non-decision time (*Ter*), and starting-point variability (*sz*).

## Starting point dynamic DDM

Here, the state of the diffusion variable was initialized according to *Equation 8*. Thus, the starting point model had 6 free parameters (the five basic ones plus the starting point bias, $z_{bias}$).

## Drift-bias dynamic DDM

The diffusion variable at time 0 was initialized according to *Equation 8*. Also, the diffusion variable in the viewing period was not updated according to *Equation 9* but according to:

$$\mathrm{dy(t)} = (\mathrm{g} \cdot M_t + v_{bias} \cdot prev) \cdot \mathrm{dt} + \mathrm{cdW} \tag{15}$$

The drift-bias model had the five basic parameters plus the drift-bias parameter ($v_{bias}$). Finally, the

hybrid dynamic DDM had two biasing parameters ($z_{bias}$ and $v_{bias}$) and overall seven free parameters. The diffusion variable was initialized according to *Equation 8* and evolved in the viewing period according to *Equation 12* and in the post-stimulus period according to *Equation 13*.

## Leaky Accumulator Models – General

The leaky accumulator model was based on models described before (*Busemeyer and Townsend, 1993*; *Zhang and Bogacz, 2010*), constituting an extension of the DDM:

$$dy = (s \cdot v + \lambda \cdot y)dt + cdW \tag{16}$$

where the rate of change of $y$ now also depends on its current value, with a magnitude controlled by the additional parameter $\lambda$, the effective leak which reflects the time constant of the accumulation process.

We defined three dynamic variants (c.f. dynamic DDM above) of the leaky accumulator model in order to account for history biases. These different biasing mechanisms were further crossed with two different bound regimes: static or collapsing bounds, as described for the DDM above.

## Leaky Accumulator with Starting Point Bias

Here, the diffusion variable was initiated according to *Equation 8*. During the viewing period, it was updated according to:

$$dy(t) = (\lambda \cdot y(t) + g \cdot Mt) \cdot dt + cdW \tag{17.1}$$

After stimulus offset, accumulation continued according to:

$$dy(t) = \lambda \cdot y(t) + \frac{y(T)}{T} + cdW \tag{17.2}$$

## Leaky Accumulator with Input Bias

Here, the diffusion variable was initiated according to *Equation 6*. The evolution of the decision variable during the viewing period was described by:

$$dy(t) = (\lambda \cdot y(t) + g \cdot Mt + v_{bias} \cdot prev) \cdot dt + cdW \tag{18}$$

After stimulus offset accumulation continued according to *Equation 17.2*. Responses were determined by a static threshold crossing mechanism, as in the standard DDM models described above.

The third leaky accumulator model we defined, the $\lambda$-*bias* model, accounted for history biases by introducing an asymmetry in the dynamics of evidence accumulation. In this model, we followed a different implementation in order to enable biasing the effective leak ($\lambda$) parameter: we reformulated the model to describe two separate accumulators that integrate the sensory evidence. We define the diffusion variable as $y = y_A - y_B$, with $y_A$ and $y_B$ being two independent accumulators coding for the right and left choice. The two accumulators were initialized as follows:

$$y_A(0) = U(-sz, sz) \tag{19.1}$$

$$y_B(0) = 0 \tag{19.2}$$

Starting point variability was thus applied only to one accumulator, which was equivalent to applying this variability on their difference (diffusion variable *y*).

During the viewing period the two accumulators were updated according to:

$$dy_A(t) = [\lambda_A \cdot y_A(t) + g \cdot f_A(M_t)] \cdot dt + \frac{cdW}{\sqrt{2}} \tag{20.1}$$

$$dy_B(t) = [\lambda_B \cdot y_B(t) + g \cdot f_B(M_t)] \cdot dt + \frac{cdW}{\sqrt{2}} \tag{20.2}$$

The variance of the processing noise applied to each accumulator was divided by two such as the processing variance of the accumulators' difference (variable $y$) is $c^2$, as in the DDM.

The functions $f_A$ and $f_B$ were threshold linear functions, with $f_A$ setting negative values to 0 and $f_B$ setting positive values to 0. Specifically:

$$f_A(x) = \begin{cases} x, & \text{if } x > 0 \\ 0, & \text{if } x \leq 0 \end{cases} \tag{20.3}$$

$$f_B(x) = \begin{cases} 0, & \text{if } x > 0 \\ -x, & \text{if } x \leq 0 \end{cases} \tag{20.4}$$

Thus, the $y_A$ accumulator 'listened' only to the negative values of the input stream while the $y_B$ only to positive values. The effective leak parameters for each accumulator were defined as follows:

$$\lambda_A = \lambda + f_A(prev) \cdot \lambda_{bias} \tag{20.5}$$

$$\lambda_B = \lambda + f_B(prev) \cdot \lambda_{bias} \tag{20.6}$$

## Leaky Accumulator with Static Bounds

A response was initiated when the difference between the two accumulators ($y$) exceeded a positive threshold +a (right choice) or surpassed a negative threshold –a (left choice). These leaky accumulator models had one biasing parameter each as well as the following five basic parameters: threshold value (a), effective leak ($\lambda$), noise scaling (g), non-decision time ($Ter$), and starting-point variability ($sz$).

## Leaky Accumulator with Collapsing Bounds

We implemented versions of the leaky accumulator models described above using collapsing bounds. For the input and starting point bias models, the time-varying bounds are described in *Equations 14.1 and 14.2*. For the $\lambda$ bias model, collapsing bounds had the same functional form but their asymptote was set to 0 (mirroring the fact that in this model the neutral point of the $y = y_A - y_B$ decision variable was at 0, rather than at a/2 as in all other models involving a single accumulator):

$$b_{up}(t) = \left| a - a \frac{t}{t+c} \right|_0^a \tag{21.1}$$

$$b_{down}(t) = \left| a \frac{t}{t+c} - a \right|_{-a}^0 \tag{21.2}$$

## Model fitting procedures

We fit the extended models using a Quantile Maximal Likelihood (QMPE) approach. Under this approach, empirical RT values are classified into bins defined by the 0.1, 0.3, 0.5, 0.7 and 0.9 quantiles of the RT distribution (six bins overall). RT quantiles were derived separately for the various coherence levels. We excluded the 81% coherence trials and pooled together the 0% and 3% coherence trials as RT quantiles in these trials were not distinguishable. This resulted in quantiles for each of three difficulty levels (0% and 3%, 9% and 27%), for each of the two responses (correct/error), and for two history conditions (motion direction in current trial *consistent* or *inconsistent* with the previous response), leading to 6 bins x 3 coherence x 2 response x two history = 72 bins per participant. Denoting the number of empirical observations in a particular bin $k$ by $n_k$ and the probability predicted by the model to derive a response in a particular bin $k$ by $P_k$, the likelihood $L$ of the data given the model is defined as:

$$L = \prod_k P_k^{n_k} \tag{22}$$

We applied a commonly used multi-stage approach to fit our simulation-based models (e.g. *Teodorescu et al., 2016*). First, each fitting session started by generating 20 random parameter sets, drawn from a uniform distribution bounded by the range of each parameter. To improve the

precision of likelihood estimates, we generated 10 synthetic trials for each experimental trial, replicating the trials for a given participant. We then computed the likelihood of the model parameters given the data. The parameter set with the best fit out of the initial 20 was used as the starting point for a standard optimization routine (*fminsearchbnd* function in Matlab, which implements a constrained version of the Nelder-Mead simplex algorithm). In total, we ran 50 of such fitting sessions, each with a different random seed. Second, we chose the best-fitting parameter set from each of the 50 sessions and recomputed the likelihood while replicating 20 synthetic trials for each experimental trial. Third, the five best-fitting of these 50 sets were used as starting points fminsearchbnd, which further refined the local minima of the fit. Fourth, we recalculated the likelihood of the single best parameter set in simulations with 30 synthetic trials for each experimental trial (see *Equations 6*). For each model $f$, AIC values were calculated at the group level:

$$AIC_f = -2\sum_{S}^{N} \ln(\mathcal{L}_s) + 2m_f \tag{23}$$

where $N$ is the total number of participants and $s$ is the participants index. $L_s$ denotes the maximum likelihood estimate for each participant. Finally, $m_f$ is the number of free parameters for a given model $f$.

## Effective bias signal

We calculated the effective bias signal (as in *Hanks et al., 2011*) for the winning leaky accumulator model with collapsing bounds (*Figure 7d*). We assumed that the current choice is biased in the direction of the previous choice (repetition bias). We arbitrarily set the previous choice to 'right' (prev = 1), which means that the biasing mechanisms pushes the decision variable closer to the upper boundary. In both models, the effective bias signal at time $t$ was obtained by dividing the value of the *cumulative bias* signal by the value of the upper bound on that moment.

We took the average of the absolute input bias parameter, so as to emulate a repetition bias. Participants were divided in two groups based on the sign of the fitted parameter λ. We calculated the effective bias signal in two instances: a) by averaging parameters across participants with λ > 0, and b) by averaging parameters across participants with λ < 0. Because the time courses were very similar in these two cases, in *Figure 6d* we show the average of the two effective bias signals.

## Model simulations

We simulated various biasing mechanisms within the frameworks of the DDM and the leaky accumulator models. Per biasing mechanism, we simulated 100K traces in timesteps of 10 ms using *Equations 2* (DDM) and *Equation 18* (leaky accumulator).

For the DDM simulations (*Figure 7—figure supplement 3*), the main parameters were: boundary separation = 1; drift rate = 1; non-decision time = 0.1; starting point = 0.5 (expressed as a fraction of the boundary separation); drift bias = 0; drift rate variability = 0.5. We simulated three levels of starting point bias (0.56, 0.62 and 0.68), three levels of constant drift bias (0.2, 0.5 and 0.8), three levels of a time-dependent linear increase in drift bias (1.5/s, 2.5/s and 3.5/s), three levels of constant drift bias (0.2, 0.5 and 0.8) in combination with hyperbolically collapsing bounds (given by *Equation 16* and using c = 3), and three levels of one time-dependent collapsing and one expanding bound: 0.2/s, 0.5/s and 0.8/s.

For the leaky accumulator simulations (*Figure 7—figure supplement 2*), the main parameters for each accumulator were: input = 1; boundary = 0.42; λ = -2.5; starting point = 0; input bias = 0. The negative λ's determined that the accumulators were self-excitatory in nature (as opposed to leaky). We choose this to match the primacy effects observed in the data (*Figure 7—figure supplement 1d*). We simulated three levels of starting point bias (0.05, 0.10 and 0.15), three levels of input bias (0.2, 0.5 and 0.8), and three levels of λ-bias between the two accumulators: (-3 vs -2, -4 vs -1, and -5 vs 0).

We then fit DDM models separately to each of the simulated datasets and fit the parameters boundary separation, drift rate, non-decision time, starting point, drift bias and drift rate variability.

## Statistical tests

We quantified across-subject correlations between P(repeat) and the individual history components in DDM bias parameter estimates using Spearman's rank correlation coefficient $\rho$. The qualitative pattern of results does not depend on the choice of a specific correlation metric. Even though individual subject parameter estimates are not independent due to the hierarchical nature of the HDDM fit, between-subject variance in parameter point estimates can reliably be correlated to an external variable - in our case, P(repeat) - without inflation of the false positive rate (*Katahira, 2016*). The difference between two correlation coefficients that shared a common variable, and its associated p-value, was computed using Steiger's test (*Steiger, 1980*).

We used Bayes factors to quantify the strength of evidence across our different datasets. We first computed the Bayes factor for each correlation (between P(repeat) and the history shift in starting point, and between P(repeat) and the history shift in drift bias) (*Wetzels and Wagenmakers, 2012*). We then multiplied these Bayes factors across datasets to quantify the total evidence in favor or against the null hypothesis of no correlation (*Scheibehenne et al., 2016*). $BF_{10}$ quantifies the evidence in favor of the alternative versus the null hypothesis, where $BF_{10} = 1$ indicates inconclusive evidence to draw conclusions from the data. $BF_{10} < 1/10$ or $> 10$ is taken to indicate substantial evidence for $H_0$ or $H_1$ (*Kass and Raftery, 1995*).

## Data and code availability

All behavioral data, model fits and analysis code are available under a CC-BY 4.0 license at https://doi.org/10.6084/m9.figshare.7268558. Analysis code is also available on GitHub (https://github.com/anne-urai/2018_Urai_choice-history-ddm; copy archived at https://github.com/elifesciences-publications/2018_Urai_choice-history-ddm; *Urai and de Gee, 2019*).

## Acknowledgements

We thank Gilles de Hollander and Peter Murphy for discussion. Anke Braun kindly shared behavioral data of the Visual motion 2AFC (FD) study. Christiane Reißmann, Karin Deazle, Samara Green and Lina Zakarauskaite helped with participant recruitment and data acquisition for the Visual motion 2IFC (FD) #2 study.

This research was supported by the German Academic Exchange Service (DAAD, to AEU), the EU's Horizon 2020 research and innovation program (under the Marie Skłodowska-Curie grant agreement No 658581 to KT) and the German Research Foundation (DFG) grants DO 1240/2–1, DO 1240/3–1, SFB 936/A7, and SFB 936/Z1 (to THD). We acknowledge computing resources provided by NWO Physical Sciences.

## Additional information

### Funding

| Funder | Grant reference number | Author |
|---|---|---|
| German Academic Exchange Service London | A/13/70362 | Anne E Urai |
| Deutsche Forschungsgemeinschaft | DO 1240/2-1 | Tobias H Donner |
| Deutsche Forschungsgemeinschaft | DO 1240/3-1 | Tobias H Donner |
| Deutsche Forschungsgemeinschaft | SFB 936/A7 | Tobias H Donner |
| Deutsche Forschungsgemeinschaft | SFB 936/Z1 | Tobias H Donner |
| H2020 Marie Skłodowska-Curie Actions | 658581 | Konstantinos Tsetsos |

The funders had no role in study design, data collection and interpretation, or the decision to submit the work for publication.

## Author contributions

Anne E Urai, Conceptualization, Data curation, Software, Formal analysis, Investigation, Visualization, Methodology, Writing—original draft, Writing—review and editing; Jan Willem de Gee, Konstantinos Tsetsos, Formal analysis, Writing—review and editing; Tobias H Donner, Conceptualization, Resources, Supervision, Writing—original draft, Writing—review and editing

## Author ORCIDs

Anne E Urai ⓘ https://orcid.org/0000-0001-5270-6513
Jan Willem de Gee ⓘ https://orcid.org/0000-0002-5875-8282
Konstantinos Tsetsos ⓘ https://orcid.org/0000-0003-2709-7634
Tobias H Donner ⓘ https://orcid.org/0000-0002-7559-6019

## Ethics

Human subjects: All participants gave written informed consent, and consent to publish. The ethics committees of the University of Amsterdam (Psychology Department), University Medical Center Hamburg-Eppendorf (PV4714), and Leiden University (Cognitive Psychology department) approved the study procedures.

## Decision letter and Author response

Decision letter https://doi.org/10.7554/eLife.46331.032
Author response https://doi.org/10.7554/eLife.46331.033

# Additional files

## Supplementary files

• Transparent reporting form
DOI: https://doi.org/10.7554/eLife.46331.020

## Data availability

All data is available at https://doi.org/10.6084/m9.figshare.7268558, and analysis code at https://github.com/anne-urai/2018_Urai_choice-history-ddm (copy archived at https://github.com/elifesciences-publications/2018_Urai_choice-history-ddm).

The following dataset was generated:

| Author(s) | Year | Dataset title | Dataset URL | Database and Identifier |
|-----------|------|---------------|-------------|-------------------------|
| Urai AE, de Gee JW, Tsetsos K, Donner TH | 2018 | Choice history biases subsequent evidence accumulation | https://doi.org/10.6084/m9.figshare.7268558 | figshare, 10.6084/m9.figshare.7268558 |

The following previously published datasets were used:

| Author(s) | Year | Dataset title | Dataset URL | Database and Identifier |
|-----------|------|---------------|-------------|-------------------------|
| Murphy PR, Vande-kerckhove J, Nieu-wenhuis S | 2014 | Pupil-linked arousal determines variability in perceptual decision making | https://doi.org/10.5061/dryad.tb542 | Dryad Digital Repository, 10.5061/dryad.tb542 |
| Urai AE, Braun A, Donner TH | 2017 | Pupil-linked arousal is driven by decision uncertainty and alters serial choice bias | http://dx.doi.org/10.6084/m9.figshare.4300043 | figshare, 10.6084/m9.figshare.4300043 |
| de Gee JW, Kna-pen T, Donner TH | 2014 | Decision-related pupil dilation reflects upcoming choice and individual bias | https://doi.org/10.6084/m9.figshare.4806559 | figshare, 10.6084/m9.figshare.4806559 |
| de Gee JW, Colizoli O, Kloosterman NA, Knapen T, | 2017 | Dynamic modulation of decision biases by brainstem arousal systems | https://doi.org/10.6084/m9.figshare.4806562 | figshare, 10.6084/m9.figshare.4806562 |

Nieuwenhuis S,
Donner TH

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
