## [Decision Letter]

Thank you for submitting your article "Choice history biases subsequent evidence accumulation" for consideration by *eLife*. Your article has been reviewed by three peer reviewers, including Timothy Verstynen as the Reviewing Editor and Reviewer #3, and the evaluation has been overseen by Barbara Shinn-Cunningham as the Senior Editor.

The reviewers have discussed the reviews with one another and the Reviewing Editor has drafted this decision to help you prepare a revised submission.

Summary:

This manuscript by Urai and colleagues reports a thorough analysis of how choice history impacts decision processes during perceptual decision-making. Analyzing data from 6 experiments (4 previously published), the authors asked whether choice history modifies the parameters of the drift diffusion model in a reliable and meaningful way. They consistently find that the influence of past choices on future decisions is best explained by a bias in drift rate toward the previously selected decision (regardless of accuracy). While there was evidence of a minor influence of history on the starting point bias, this effect was largely dwarfed by the modulation of the drift rate. The authors contrast their results with prior work suggesting that choice history primarily shifts the starting point of the decision process

This is an extremely well-written paper, supported by a sound, thorough, and sophisticated set of analyses and strikingly clear visualization. The authors' dedication to replicating the computational mechanism is laudable, and the (five-times-replicated) results give strongly convincing evidence that choice history biases the drift-rate in perceptual decision-making. That being said, there are several open questions that remain unresolved.

All reviewers were enthusiastic about the paper, however, they also identified several issues that need to be addressed before the study can be accepted.

Essential revisions:

1) Dual parameter models. All three reviewers thought that there needed to be a more in depth investigation of the dual parameter models.

Reviewer 1 pointed out that the authors repeatedly conclude that history shifts in drift rate, but not starting point, account for choice history effects, but they show that a model that lets both of these parameters vary with choice history fits better than either one alone in 5 out of 6 datasets. This is potentially interesting, and the paper would benefit from a more nuanced/accurate interpretation of this result. As it is currently written, the narrative is that either of these mechanisms could account for choice history effects, but only drift rate does. While drift rate seems to have a stronger effect, the authors need to present an intuition about why the model including both as free parameters fits the best. What does the starting point add, that the drift rate doesn't account for? How do the starting point and drift rate interact? Is it the case that one (starting point) accounts for choice history effects on fast RT trials, and the other (drift rate) accounts for effects on slower RT trials? Delving deeper here could make the paper richer. To this point, in Figure 3B, what is the performance of a model including both z and v_bias_?

Reviewer 1 also points out that, in Figure 6, for the model that incorporates trial history, it looks like only 2 of 6 datasets are best fit by a model including both z and v_bias_, and most of the rest favor v_bias_. Does this reflect their parameterization of z? Did the authors explore the possibility that z and v_bias_ might integrate over different timescales? The Materials and methods section reports that they fit the same regressors (X) for v_bias_ and z in the HDDM regression models; if that's not the case, the authors should clarify. z seems to help model performance when it reflects one trial back, why is that not the case for longer timescales?

Reviewer 2 pointed out that the authors appear to treat choice history effects as monolithic with respect to mechanism. That is, they test whether influences of previous choices are universally (across the tasks explored) accounted for by certain model parameters and not others, whether those relate to response repetition versus alternation. This assumption may ultimately be justified, but the paper does not offer support for it from the literature or from the data examined. It would be appropriate to include a set of analyses that split the drift and starting point biases in two, with one set of fit parameters increasing the bias towards a given response after it is chosen, and the other set decreasing that bias. (The authors should select whether to do this in the standard DDM framework or one of their better-performing models, but they certainly do not need to test this across their wide set of models.) How does such a model compare to their current version that does not distinguish between these response biases? Do the currently observed relative influences of drift and starting point hold across both directions of bias?

Finally, reviewer 3 pointed out that the original model comparisons (Figure 3) show that in 4 of the 6 experiments a dual parameter model best explains behavior. While the authors make a clear case that choice bias drives drift-rate bias changes, it could be that another factor (e.g., accuracy) is also modifying starting point bias. In fact, the previous papers cited above show that accuracy biases boundary height. Given this previous literature, the authors should consider what other factors may independently be manipulating starting point bias (and boundary height).

2) Choice history effects. All three reviewers requested a more detailed elaboration of the nature of the choice history effects.

Reviewer 1 pointed out that, in Figure 4, when the authors compare history shifts in z and v_bias_ to each subjects' choice repetition, are they only looking at trials where the current evidence and the previous choice were congruent (meaning, if I chose right on trial t-1, and right evidence > left evidence on trial t)? If so, this should be clarified in the text. If not, is this why the z parameter does so poorly? How would these plots look for trials when previous choice and current evidence are congruent vs. incongruent?

Reviewer 2 strongly recommended revising the Introduction and the start of Results to enumerate the choice history biases that have previously been observed, and the approach of the current analyses in clustering these together (this was not clear on first read).

Reviewer 2 also pointed out that it wasn't clear what distinction the authors were drawing between the prior reported effects and the current experiments – they say that the current ones "emerge spontaneously and in an idiosyncratic fashion" but estimates of prior probability could also change spontaneously and idiosyncratically. Perhaps the distinction they are drawing (alluded to in the following paragraph) is between environments that naturally engender a rational basis for choice history biases (e.g., autocorrelation between responses) versus those that do not. This could still predict that the same rational biases that produce starting-point biases in other experiments could result in shorter term history effects in experiments like the current one, if participants treat short runs of the same response as weak evidence for autocorrelation. The current findings suggest that this is not the case for these data. Also it is unclear whether the authors find any cumulative effects of choice history if they look at how many times a given response was given prior to the current response (this is similar to the current lagged analyses but assumes a specific cumulative effect).

Reviewer 3 pointed out that the authors' lack evidence to justify their claim that the accuracy of the preceding trials does not influence choice history effects on the drift rate. This inference is largely based on the fact that the drift rate correlations with repetition probability pass a significance threshold (Bayes factor and p-value) both when the previous trial was correct and when it was incorrect (Figure 5). However, the magnitude of the likelihood ratios (i.e., BF_10_'s) look consistently smaller for previous error trials than previous correct trials. No direct comparison on the magnitude of these effects is run, thus the actual null hypothesis as to whether they are different was never evaluated.

3) Alternative models. Both reviewers 2 and 3 had concerns about possible alternative mechanisms.

Reviewer 2 pointed out that the authors currently interpret choice history as being causal on changes in drift across trials, but they do not address potential third variables that could commonly influence both. For instance, could autocorrelated changes in drift rate across trials (due to some unrelated generative function, e.g., slowly drifting over the experiment) drive the observed choice history biases? There isn't a strong intuition that this should be the case but it is a plausible mechanism (i.e., as an autocorrelated alternative to typical formulations of drift variability) and it would be easy enough to simulate this in order to rule it out. Similarly, could an alternate form of autocorrelation that reflects regression-to-the-mean on drift rate (i.e., larger drift rates are more likely to be followed by smaller ones) produce these biases?

Reviewer 3 pointed out that, several papers have looked at how reinforcement learning mechanisms target specific parameters of accumulation-to-bound processes (see Frank et al., 2015; Pedersen et al., 2017; Dunovan and Verstynen, 2019). The critical difference between these studies and the current project is that all three found evidence that selection accuracy targets the boundary height parameter itself. Not only is it relevant to link the current study to this previous work, but it also begs the question as to why the authors did not test the boundary height parameter as well. The motivation for comparing models featuring starting point and drift-rate bias terms, respectively, makes sense. While the current analyses point to an accuracy-independent effect of choice bias (see point #2 for comments on accuracy analysis), given the previous findings showing the dependence between of the boundary height and selection accuracy a model testing this effect should be considered. Alternatively, given that the decision boundary is a relevant parameter and accuracy has been shown to modulate the boundary, the authors should give their rationale for its exclusion, in combination with a convincing response to the critique of the accuracy analysis (see below).

Frank MJ, Gagne C, Nyhus E, Masters S, Wiecki TV, Cavanagh JF, et al. fMRI and EEG predictors of dynamic decision parameters during human reinforcement learning. J Neurosci. 2015;35(2):485-494

Pedersen ML, Frank MJ, Biele G. The drift diffusion model as the choice rule in reinforcement learning. Psychonomic bulletin and review. 2017;24(4):1234-1251.

Dunovan K, and Verstynen T. "Errors in action timing and inhibition facilitate learning by tuning distinct mechanisms in the underlying decision process." J Neurosci (2019): 1924-18.

4) Leaky competing accumulators. Reviewer 3 had concerns about the LCA simulations. The extension of the findings to models with collapsing bounds is interesting as a robustness analysis. However, the incorporation of the LCA model seems less straightforward. If input bias is the LCA parameter most similar to the drift rate in the DDM, then what are we to make of model 4 (LCA without collapsing bounds) that shows λ-bias being the best model (Figure 7B)? It is completely left out of Figure 7C. All in all, this analysis seemed to simply "muddy the water" on the main results.

5) Mechanism. Reviewer 3 also requested more details on the proposed mechanism. The idea that changes in a history-dependent drift-rate bias correspond to shifts in endogenous attention from one interpretation to another could be further developed. In particular it seems to imply a more explicit (or at least non-procedural) mechanism for trial-wise changes in drift rate bias. But one could imagine many other mechanisms driving this effect as well (see papers referenced above). As it stands, the connection between shifts in attention and drift-rate bias does seem to be an intuitively plausible explanation (one of many), but a further description of the authors' line of thought would help to convince the reader.

---

## [Author Response]

Essential revisions:1) Dual parameter models. All three reviewers thought that there needed to be a more in depth investigation of the dual parameter models.Reviewer 1 pointed out that the authors repeatedly conclude that history shifts in drift rate, but not starting point, account for choice history effects, but they show that a model that lets both of these parameters vary with choice history fits better than either one alone in 5 out of 6 datasets. This is potentially interesting, and the paper would benefit from a more nuanced/accurate interpretation of this result. As it is currently written, the narrative is that either of these mechanisms could account for choice history effects, but only drift rate does. While drift rate seems to have a stronger effect, the authors need to present an intuition about why the model including both as free parameters fits the best.

Thank you for raising this important point. We reply to each of your specific questions pertaining to this issue in the following. Upfront, we did not intend to claim that only drift bias “accounts for choice history effects” in a general sense. What we do want to claim (and what we think is strongly supported by our data) is more specific: that history-dependent shifts in drift bias, not starting point, explain individual differences in overt choice repetition behavior. Even so, the previous choice consistently shifts the average starting point towards negative values. We now realize that the latter aspect was not sufficiently reflected in our previous presentation of the results. We have now elaborated on the starting point effects by means of (i) additional analyses and (ii) a new paragraph in the Discussion section.

Specifically, we added the following new analyses (described in more detail below):

- Statistical tests of the average history shift in starting point (Figure 4—figure supplement 5A);

- The relationship between RT and overall repetition bias (rather than individual choice history bias; Figure 4—figure supplement 5C);

- The average correlation between individual drift bias and starting point estimates for each dataset (subsection “History-dependent accumulation bias, not starting point bias, explains individual differences in choice repetition behaviour”, last paragraph);

- Simulations of various DDM and leaky accumulator models, and DDM fits of these synthetic data (Figure 7—figure supplement 2 and 3).

Furthermore, we added the following Discussion paragraph:

“While we found that choice history-dependent variations of accumulation bias were generally more predictive of individual choice repetition behavior, the DDM starting point was consistently shifted away from the previous response for a majority of participants (i.e., negative values along x-axis of Figure 4A). […] Future work is needed to illuminate this issue, for example through manipulating decision speed and/or the delays between subsequent motor responses, and modeling choice-related neural dynamics in the motor cortex.”

What does the starting point add, that the drift rate doesn't account for?

Please see the above Discussion paragraph for our general take on this issue. The starting point shift might either reflect a “real mechanism” (but with little impact on individual choice repetition behavior), or actual limitations of the standard DDM. To explore the second possibility, we have extended our simulations of bounded accumulation models. Figure 7—figure supplement 2 and 3 now show, for each model:

1) The conditional bias function when simulating data with varying bias strength;

2) The recovered parameter estimates for z and v_bias_, when fitting a hybrid model;

3) DBIC values from a model without history for z-only, v_bias_-only, and the hybrid model;

4) Correlations between synthetic individuals’ P(bias), and the parameter estimates for z and v_bias_, as estimated from the hybrid models.

This analysis reveals two specific mechanisms that can produce both the lowest BIC for a hybrid model, together with an opposite effect on DDM drift bias and starting point:

1) A nonlinearly increasing dynamic bias (i.e. ramping drift bias) in an extended perfect accumulator model (Figure 7—figure supplement 2);

2) A leak bias in an accumulator model (Figure 7—figure supplement 3), to which we point in the Discussion paragraph (quoted above).

How do the starting point and drift rate interact?

We used two approaches to illuminate this question. First, we correlated the parameter estimates for history shift in starting point with the history shift in drift bias across observers. Correlations were generally negative (mean Spearman’s rho: -0.2884, range -0.4130 to 0.0757), the correlation coefficient only reached significance at p < 0.05 in one dataset. The combined Bayes Factor BF_10_ was 0.0473, indicating strong evidence for H_0_ (Wetzels and Wagenmakers, 2012). This is now reported in the section "History-dependent accumulation bias, not starting point bias, explains individual differences in choice repetition behavior”. Second, we used simulations (Figure 7—figure supplement 2 and 3) to unravel which modes of decision dynamics may in principle give rise to such a negative correlation: this pattern can arise from a non-linearly increasing dynamic bias (i.e., ramping drift bias Figure 7—figure supplement 2, third column), or from a leaky accumulation process with a bias in effective leak (Figure 7—figure supplement 3, third column).

Is it the case that one (starting point) accounts for choice history effects on fast RT trials, and the other (drift rate) accounts for effects on slower RT trials? Delving deeper here could make the paper richer.

This is a very interesting idea, which we now elaborate on in the Discussion paragraph. All features of the current results may be explained by the combined effects two distinct biasing mechanisms, whose effects superimpose, but contribute differently to fast and slow choices: (i) motor response alternation which will load on starting point and dominate fast choices; and (ii) an accumulation bias, which will load on drift bias and dominate slow choices.

Further, the starting point effect may be stereotypical (e.g. hardwired in the motor machinery), which is why it is generally negative for most subjects as we found in our DDM fits. Such a mechanism may be explained by the selective “rebound” of beta-band activity in the motor cortex (e.g., Pape and Siegel, 2016). By contrast, the drift bias may be more adaptive, varying with subjects’ belief about the structure of the environment, and thus giving rise to the individual differences in repetition behavior we observe in our participants overall.

In all our data sets, RTs are long so that the drift bias would be expected to dominate repetition behavior, just as we found. However, this scenario makes specific predictions when sorting trials based on RT: at very short RTs, we should predominantly find choice alternation, which is invariable across all participants, due to the stereotypical starting point shift. Furthermore, the starting point shift should only be caused by the immediately preceding response, whereas the drift bias may be affected by several choices back into the past.

In Figure 4—figure supplement 5C, we now test these predictions. This shows the overall probability of choice repetition as a function of RT quantile (not correcting for individual differences in choice history bias, as we do throughout the main figures). As expected, for the dataset which including very short RTs (< 600ms), we do observe a strong choice alternation bias on average across the group. Since not all of our datasets have many trials with such short RTs, we do not want to make too strong claims about this pattern and placed the RT split analysis in the supplement. Future work should systematically test this scenario, for example through manipulating decision speed and/or the delays between subsequent motor responses, and by modeling decision-related neural dynamics in the motor cortex.

To this point, in Figure 3B, what is the performance of a model including both z and v_bias_?

The model prediction for the hybrid model CBF is now added into the new Figure 3B. It is overall quite similar to the v_bias_-only model.

Reviewer 1 also points out that, in Figure 6, for the model that incorporates trial history, it looks like only 2 of 6 datasets are best fit by a model including both z and v_bias_, and most of the rest favor v_bias_. Does this reflect their parameterization of z?

This is a very perceptive point. Indeed, we also noticed and explored this discrepancy, and concluded that it is due to differences in the model fitting approaches for the two analyses – specifically between standard DDM fits used for all main analyses reported in this paper, and the regression approach used to Figure 6 (Wiecki et al., 2013). The differences persist even after fitting the regression model analogously to the fits in the other figures – i.e. only based on the choice (not correct and incorrect choice) for only lag 1 (data not shown). We believe this is a technical issue, which considers attention by the field of sequential sampling models, but is beyond the scope of the present paper. We have here used the standard fitting approach for all analyses in this paper except for the analysis in Figure 6, which can only be performed with the regression approach. We feel more comfortable to base our selection between competing models on the standard approach in general.

The purpose of the comparisons of different regression models is more methodological in nature: to find the best-fitting lag across trials. So, we have now moved this panel from main Figure 6 to Figure 6—figure supplement 1, instead focusing main Figure 6 on a conceptually more informative issue: the differences in the timescales of the history effects on drift and starting point (see reply below).

Did the authors explore the possibility that z and v_bias_ might integrate over different timescales?

Thank you for this very interesting suggestion. We now assess this quantitatively in the main Figure 6A. We fit exponentials to these “history kernels” in order to estimate the timescale of drift bias and starting point effects. Interestingly, the time constant for starting point is lower than the one for drift bias, in particular after error trials. This difference further speaks to the mechanistically distinct nature of these two history effects. We point out this difference in timescale in the Results section (subsection “Accumulation bias correlates with several past choices”) and again refer to it in Discussion.

The Materials and methods section reports that they fit the same regressors (X) for v_bias_ and z in the HDDM regression models; if that's not the case, the authors should clarify.

We realized that our Materials and methods section was not sufficiently clear, thank you for pointing this out. We have now clarified our description of the approach (section ‘HDDM regression models’):

“We first created a matrix 𝑋, with dimensions trials x 2 * lags, which included pairs of regressors coding for previous stimuli and choices (coded as −1,1), until (and including) each model’s lag. […] Starting point was defined asz~1+Xz, with a logistic link function 11+e-X.”

z seems to help model performance when it reflects one trial back, why is that not the case for longer timescales?

We realize that our model comparison visualization did not reflect the results sufficiently clearly. We have now improved the visualization, which shows that there is a consistent relationship between the three models: in those datasets where the hybrid model has the lowest AIC, this is true across several lags (up to lag 2 or 4).

See also our reply above (sixth response to point #1): these regression model comparisons are not important conceptually (only used to select a particular lag), and distract from the more important message on the timescale of integration. Thus, we have moved the full set of model comparison indices to Figure 6—figure supplement 1.

Reviewer 2 pointed out that the authors appear to treat choice history effects as monolithic with respect to mechanism. That is, they test whether influences of previous choices are universally (across the tasks explored) accounted for by certain model parameters and not others, whether those relate to response repetition versus alternation. This assumption may ultimately be justified, but the paper does not offer support for it from the literature or from the data examined. It would be appropriate to include a set of analyses that split the drift and starting point biases in two, with one set of fit parameters increasing the bias towards a given response after it is chosen, and the other set decreasing that bias. (The authors should select whether to do this in the standard DDM framework or one of their better-performing models, but they certainly do not need to test this across their wide set of models.) How does such a model compare to their current version that does not distinguish between these response biases? Do the currently observed relative influences of drift and starting point hold across both directions of bias?

In its current form, our model can (and does) capture biases either towards a previously given response (repetition), or away from the previously given response (alternation). Indeed, across our groups of observers, we see that the choice repetition probabilities of different individuals lie along a continuum from repeaters to alternators.

To investigate whether the effects on DDM parameters are the same within each of these two sub-groups of participants, we split the participants by their overall history bias: repeaters (P(repeat)>0.5) and alternators (P(repeat)>0.5). We then correlated individual repetition probability with each of the history-shifts in each of the two DDM parameters, separately within each sub-group. This was only possible for five out of the six datasets, where there were sufficient numbers of participants in both sub-groups (Figure 2B). The main result holds also within each sub-group: history shift in drift bias, not starting point, explains individual differences in choice repetition probability. This further corroborates the notion that we can treat individual choice history bias as lying on a single continuum, rather than arising from two qualitatively distinct mechanisms between these two sub-groups of individuals, or behavioral patterns.

This result is now reported in the text as follows:

“The same effect was present when individual participants were first split into “Repeaters” and “Alternators” based on P(repeat) being larger or smaller than 0.5, respectively (Figure 4—figure supplement 3)”. We hope this gets to your point.

Finally, reviewer 3 pointed out that the original model comparisons (Figure 3) show that in 4 of the 6 experiments a dual parameter model best explains behavior. While the authors make a clear case that choice bias drives drift-rate bias changes, it could be that another factor (e.g., accuracy) is also modifying starting point bias. In fact, the previous papers cited above show that accuracy biases boundary height. Given this previous literature, the authors should consider what other factors may independently be manipulating starting point bias (and boundary height).

As you suggested, we observe that the previous trial’s correctness affects both boundary separation and overall drift rate. This phenomenon of post-error slowing, and its algorithmic basis in the DDM, is present in some of our datasets (Figure 4—figure supplement 4). However, the results are not highly consistent across datasets, most likely reflecting the fact that our tasks were not conducted under speed pressure, and featured considerable sensory uncertainty.

In general, the reviews have indicated to us that we have not sufficiently addressed the link between our current work on choice history biases and previous work on other forms of sequential effects in decision-making, in particular post-error slowing. Correspondingly, we have now further unpacked the conceptual differences between these two distinct forms of sequential effects in the Introduction.

Additionally, we have now added an analysis where we allow both post-error slowing (i.e. previous correctness affecting bound height and overall drift rate) and choice history bias (i.e. previous choice affecting starting point and drift bias). These two processes seem to be relatively independent in these data: the joint fit shows largely similar results of post-error slowing (compare Figure 4—figure supplement 4B, C with D, E) as well as choice history bias (Figure 4—figure supplement 4F).

2) Choice history effects. All three reviewers requested a more detailed elaboration of the nature of the choice history effects.

Thank you for this suggestion. We realize that it was not sufficiently clear how our current approach and results differ from the previous work on sequential effects in decision making – specifically from previous work on post-error slowing. We have now changed the Introduction as follows:

“Decisions are not isolated events, but are embedded in a sequence of choices. Choices, or their outcomes (e.g. rewards), exert a large influence on subsequent choices (Sutton and Barto, 1998; Sugrue et al., 2004). […] Choice history biases vary substantially across individuals (Abrahamyan et al., 2016; Urai et al., 2017).”

Results:

“Within the DDM, choice behavior can be selectively biased toward repetition or alternation by two mechanisms: shifting the starting point or biasing the drift towards (or away from) the bound for the previous choice (Figure 1). […] History dependent changes in bound separation or mean drift rate may also occur, but they can only change overall RT and accuracy: those changes are by themselves not sufficient to bias the accumulation process toward one or the other bound, and thus towards choice repetition or alternation (see Figure 4—figure supplement 4).”

Reviewer 1 pointed out that, in Figure 4, when the authors compare history shifts in z and v_bias_ to each subjects' choice repetition, are they only looking at trials where the current evidence and the previous choice were congruent (meaning, if I chose right on trial t-1, and right evidence > left evidence on trial t)? If so, this should be clarified in the text. If not, is this why the z parameter does so poorly? How would these plots look for trials when previous choice and current evidence are congruent vs. incongruent?

Our analyses did not distinguish whether previous choice and current stimulus were congruent; all trials were included. This is now clarified in the text:

“For instance, in the left vs. right motion discrimination task, the history shift in starting point was computed as the difference between the starting point estimate for previous ‘left’ and previous ‘right’ choices, irrespective of the category of the current stimulus.”

The possibility of testing whether the previous choice induces a ‘confirmation bias’ on the next trial, increasing perceptual sensitivity to stimuli that are congruent with the previous choice, is intriguing. In fact, we have previously investigated this same question in the context of a more complex sequential decision task that focused on the interaction between two successive judgments made within the same protracted decision process (Talluri et al., 2018). In that manuscript, we describe a model-based analysis approach that is specifically designed to tackle your question. However, the 2AFC structure of current datasets is not suited for the consistency-dependent analyses used in this previous work.

Specifically, splitting trials by whether previous choices are congruent or incongruent leads to a split that is dominated by repetition vs. alternation trials. For example, in trials where the previous choice was “up” in the up/down discrimination task, and the direction of the current stimulus is also up (‘congruent’), choice repetition is correct, and choice alternation is incorrect. Similarly, for trials where the previous choice was “up” and the current stimulus motion is down (‘incongruent’), choice alternation is correct, whereas choice repetition is incorrect. Because participants chose the correct stimulus category on the majority of trials, the ‘congruent’ condition will be primarily populated by choice repetitions, whereas the ‘incongruent’ condition will be primarily populated by alternations. In sum, this analysis is confounded by participants’ above-chance performance and does not reveal any mechanism of choice history bias.

While we agree that this issue is an important direction for future research, it would require different experimental designs that are beyond the scope of our current paper. Again, we point the reviewers (and interested reader) to our recent work on confirmation bias (Talluri et al., 2018).

Reviewer 2 strongly recommended revising the Introduction and the start of Results to enumerate the choice history biases that have previously been observed, and the approach of the current analyses in clustering these together (this was not clear on first read).

This has now been addressed in the two new paragraphs (from Introduction and Results sections) quoted above.

Reviewer 2 also pointed out that it wasn't clear what distinction the authors were drawing between the prior reported effects and the current experiments – they say that the current ones "emerge spontaneously and in an idiosyncratic fashion" but estimates of prior probability could also change spontaneously and idiosyncratically.

There are two crucial ways in which choice history biases differ from previous studies of choice biases that have applied sequential sampling (“bounded accumulation”) models of decision-making (Hanks et al., 2011; Mulder et al., 2012). First, in the previous experiments, the biases were experimentally induced (through block structure in animals, or explicit task instruction or single-trial cues in humans); by contrast, no such experimental manipulation was performed in our study. (Observers were neither asked pay attention or use the past experimental sequence in any way, nor was there any sequential structure on average). Second, when prior probability was manipulated in the previous experiments, this was the probability of the occurrence of a particular target; not the conditional probability of occurrence, given a previous choice (or other experimental event). This is the difference between a frequency bias and a transition bias (see, e.g. Meyniel et al., PLoS Comput. Biol., 2016).

We have now edited the Discussion paragraph to better explain this difference:

“It is instructive to relate our results with previous studies manipulating the probability of the occurrence a particular category (i.e., independently of the sequence of categories) or the asymmetry between rewards for both choices. Most of these studies explained the resulting behavioral biases in terms of starting point shifts (Leite and Ratcliff, 2011; Mulder et al., 2012; White and Poldrack, 2014; Rorie et al., 2010; Gao et al., 2011; but only for decisions without time pressure, see Afacan-Seref et al., 2018). […] By contrast, the choice history biases we studied here emerge spontaneously and in an idiosyncratic fashion (Figure 2E), necessitating our focus on individual differences. "

Perhaps the distinction they are drawing (alluded to in the following paragraph) is between environments that naturally engender a rational basis for choice history biases (e.g., autocorrelation between responses) versus those that do not. This could still predict that the same rational biases that produce starting-point biases in other experiments could result in shorter term history effects in experiments like the current one, if participants treat short runs of the same response as weak evidence for autocorrelation. The current findings suggest that this is not the case for these data.

This distinction and the one we referred to in the quote above are different. Here, you refer to the distinction between environments with and without serial correlations in the stimuli (more precisely: stimulus categories). Let us elaborate on our assumptions about this here.

Based on insights from recent normative models (e.g. Yu and Cohen, 2009; Glaze et al., 2015) and our own empirical work (Braun et al., 2018), we suspect there is no fundamental difference between task environments that naturally engender choice history bias, and those who don’t. While idiosyncratic choice history biases appear across almost all task environments (even random ones, in which the experimenter intended to eliminate them) they are generally stronger and more consistent across people when the stimulus sequence exhibits correlation structure. In that framework of the above normative models, the existence of history biases in the face of random sequences can be readily explained by assuming that participants bring an internal representation of environmental stability to the lab that is biased toward repetition or alternation. If decision-makers assume that evidence is stable (i.e., repeating across trials), they should accumulate their past decision variables into a prior for the current trial; if they assume the evidence is systematically alternating, they should accumulate past decision variables with sign flips, yielding alternating priors (Glaze et al., 2015). Note that a stability assumption would make sense, because natural environments typically have strong autocorrelation (Yu and Cohen, 2009) just as one finds experimentally.

This is what we mean when we speculate: “… these considerations suggest that participants may have applied a rational strategy, but based on erroneous assumptions about the structure of the environment.”

Also it is unclear whether the authors find any cumulative effects of choice history if they look at how many times a given response was given prior to the current response (this is similar to the current lagged analyses but assumes a specific cumulative effect).

We investigate these cumulative effects by plotting the probability that a trial is a repetition of the previous choice, conditioned on the sequence of choices that preceded it. So, for lag 0 this equals the average P(repeat) across all trials, for lag 1 we take only those trials following a repetition, for lag 2 we take trials following two consecutive repeats, etc. for longer sequences of repetitions. The same analyses can be performed conditioning on increasingly long sequences of alternations.

We observe that, across all observers and datasets, repetition indeed increases following longer sequences of repeats. We refer to this cumulative effect as ‘cumulative repetition bias’. This cumulative bias saturates around 4 consecutive repeats, similar to effects previously reported in humans (e.g. Yu and Cohen, 2009; Cho et al., 2002; Kirby, 1976; Soetens et al., 1985; Sommer et al., 1999) and in rats (Hermoso-Mendizabal et al., 2018, their Figure 2E).

**Author response image 2. respfig2:** 

Reviewer 3 pointed out that the authors' lack evidence to justify their claim that the accuracy of the preceding trials does not influence choice history effects on the drift rate. This inference is largely based on the fact that the drift rate correlations with repetition probability pass a significance threshold (Bayes factor and p-value) both when the previous trial was correct and when it was incorrect (Figure 5). However, the magnitude of the likelihood ratios (i.e., BF_10_'s) look consistently smaller for previous error trials than previous correct trials. No direct comparison on the magnitude of these effects is run, thus the actual null hypothesis as to whether they are different was never evaluated.

Thank you for this perceptive point. We have now added direct comparison between the correlation coefficients after correct and error trials (Figure 5D). In fact, we had also discussed among ourselves prior to submission, but decided against addressing it further due to the associated additional complexity of the analysis.

For the data presented in our previous Figure 4, this comparison would be confounded, due to the asymmetries in trial counts for correct and incorrect choices. As shown in in Author response image 3, when drawing (for each synthetic subject) two random sequences of binomial coinflips with the same underlying probability, and then correlating these between synthetic subjects, the between-subject correlation decreases with fewer trials in each sequence. As an example (with n=32, as in our Visual motion 2AFC (FD) dataset), the median trial counts for error and correct are indicated on the left. This pattern does not decrease with a larger number of subjects, and reflects the added noise in the correlation coefficient when individual datapoints are less precisely estimated.

**Author response image 3. respfig3:** 

To compute the comparison between the correlation coefficients in an unbiased manner, we have now subsampled the post-correct trials to equate the number of post-error trials for each participant and dataset (throughout Figure 5). Based on Bayes factors on the difference in the resulting correlation coefficients, we are not able to strongly refute nor confirm the null hypothesis of no difference (Figure 5D). In sum, we observe a significant positive correlation between P(repeat) and drift bias after both error and correct trials, showing that qualitatively similar mechanisms are at play after both outcomes. That said, we do *not* claim that these effects are identical.

3) Alternative models. Both reviewers 2 and 3 had concerns about possible alternative mechanisms.Reviewer 2 pointed out that the authors currently interpret choice history as being causal on changes in drift across trials, but they do not address potential third variables that could commonly influence both. For instance, could autocorrelated changes in drift rate across trials (due to some unrelated generative function, e.g., slowly drifting over the experiment) drive the observed choice history biases? There isn't a strong intuition that this should be the case but it is a plausible mechanism (i.e., as an autocorrelated alternative to typical formulations of drift variability) and it would be easy enough to simulate this in order to rule it out. Similarly, could an alternate form of autocorrelation that reflects regression-to-the-mean on drift rate (i.e., larger drift rates are more likely to be followed by smaller ones) produce these biases?

To address this question, we simulated an ARIMA process that is defined by a parameter c, the auto-correlation at lag 1, and with standard deviation 0.1. We then added this fluctuating pertrial value to the drift rate, rather than using a single fixed drift rate term across trials. In Author response image 4, we allow c to vary positively from 0.1 to 1 (in steps of 0.1; blue, reflecting autocorrelated sequences) and negatively from -0.1 to -1 (in steps of 0.1; orange, reflecting ‘regression to the mean’ sequences). In neither case do we observe repetition or alternation biases, assessed based on P(repeat), nor a bias in starting point or drift. This rules out the notion that autocorrelations, or regression to the mean, on drift rates can produce any of the history-dependent bias effects we have quantified here.

**Author response image 4. respfig4:** 

This result is a specific demonstration of the general point (now elaborated in the paper), that changes in mean drift rate (or boundary separation) alone cannot produce any selective bias towards one or the other choice. The mean drift rate v is an unsigned quantity that affects the decision through its multiplication with the stimulus category s:

dy = s ∙ v ∙ dt + cdW (1)

whereby s is coded as [-1,1] to represent ‘up’ or ‘down’ stimuli. If v is reduced over time (e.g. due to fatigue), this slows down decisions and leads to more errors (and can cause biased estimates of post-error slowing, (Dutilh et al., 2012) but it won’t by itself produce more choice repetitions or alternations. Some change in either starting point or drift bias is necessary for the latter.

Now, it may be possible that a third variable causes drift bias to vary slowly over time, which will then cause choice repetition. This is a viable alternative to the one that choices bias subsequent drift, to give rise to choice repetition (or alternation). Indeed, both scenarios would give rise to autocorrelations in the (signed) drift across trials. One plausible third variable that might cause such slow changes in drift bias towards one or the other bound is selective attention (see also our reply to your point on attention below). We now elaborate on this issue in our paragraph on possible underlying mechanism in our Discussion paragraph on attention.

“It is tempting to speculate that choice history signals in these regions cause the same top-down modulation of sensory cortex as during explicit manipulations of attention. […] These ideas are not mutually exclusive and can be tested by means of multiarea neurophysiological recordings combined with local perturbations.”

We think that both scenarios are interesting, and that conclusively distinguishing between them will require interventional approaches in future work. We are aware of the limitations of our current correlational approach, and we realize that some statements in the previous version of our manuscript may have suggested a strong causal interpretation. We have now rephrased down all statements of this kind throughout the paper. Thank you for pointing us to this important issue.

Reviewer 3 pointed out that, several papers have looked at how reinforcement learning mechanisms target specific parameters of accumulation-to-bound processes (see Frank et al. 2015; Pedersen et al. 2017; Dunovan and Verstynen, 2019). The critical difference between these studies and the current project is that all three found evidence that selection accuracy targets the boundary height parameter itself. Not only is it relevant to link the current study to this previous work, but it also begs the question as to why the authors did not test the boundary height parameter as well. The motivation for comparing models featuring starting point and drift-rate bias terms, respectively, makes sense. While the current analyses point to an accuracy-independent effect of choice bias (see point #2 for comments on accuracy analysis), given the previous findings showing the dependence between of the boundary height and selection accuracy a model testing this effect should be considered. Alternatively, given that the decision boundary is a relevant parameter and accuracy has been shown to modulate the boundary, the authors should give their rationale for its exclusion, in combination with a convincing response to the critique of the accuracy analysis (see below).Frank MJ, Gagne C, Nyhus E, Masters S, Wiecki TV, Cavanagh JF, et al. fMRI 1037and EEG predictors of dynamic decision parameters during human reinforcement 1038 learning. J Neurosci. 2015;35(2):485-494Pedersen ML, Frank MJ, Biele G. The drift diffusion model as the choice rule in 1105reinforcement learning. Psychonomic bulletin and review. 2017;24(4):1234-1251.Dunovan K, and Verstynen T. "Errors in action timing and inhibition facilitate learning by tuning distinct mechanisms in the underlying decision process." J Neurosci (2019): 1924-18.

Thank you for this important point. Figure 4—figure supplement 4 shows two analyses: first (A-C), we test for the pure effect of post-accuracy increases in boundary separation (A) and reductions in mean drift rate (v) – two established sources of post-error slowing (Purcell and Kiani, 2016). While we observe post-error slowing in some of our datasets, this mainly loads onto the drift rate parameter.

We replicate these results when adding choice history bias into the same model (D-F), indicating that choice history biases (specifically via drift bias) and post-error slowing independently shape overall decision dynamics in our data.

Post-error slowing has previously been studied within the DDM framework, whereas choice history bias has not – so our focus is on the latter. Importantly, our control analyses show that all our conclusions pertaining to the mechanisms of choice history bias do not depend on whether or not post-error slowing effects are taken into account (Figure 4—figure supplement 4F).

4) Leaky competing accumulators. Reviewer 3 had concerns about the LCA simulations. The extension of the findings to models with collapsing bounds is interesting as a robustness analysis. However, the incorporation of the LCA model seems less straightforward. If input bias is the LCA parameter most similar to the drift rate in the DDM, then what are we to make of model 4 (LCA without collapsing bounds) that shows λ-bias being the best model (Figure 7B)? It is completely left out of Figure 7C. All in all, this analysis seemed to simply "muddy the water" on the main results.

Thank you for pointing this out – we agree that the two types of LCA models were confusing. We have removed model 4 from Figure 7, focusing now on the leaky accumulator models including collapsing bounds.

That said, we view the leaky accumulator model as more than just a “robustness check“. A substantial body of theoretical work (Usher and McClelland, 2001; Wong and Wang, 2006; Roxin, 2008; Brunton et al., 2013; Ossmy et al., 2013; Brunton et al., 2013; Glaze et al., 2015) indicates (i) that models with a non-zero leak term provide a more accurate description of decision dynamics and (ii) that leaky evidence accumulation is, in fact, normative under general conditions, in which the sensory evidence is *not* stationary (Ossmy et al., 2013; Glaze et al., 2015). Moreover, the leaky accumulator model is attractive in enabling the distinction between two forms of accumulation biases: a bias in the input feeding into the accumulators, or a bias in the accumulation itself. This helps generate more specific mechanistic hypotheses for future neurophysiological tests from the behavioral modelling. Similar points could be made for the collapsing bounds.

So, while we think use of the DDM is perfectly motivated from its general simplicity and wide use in the field, generalization of our conclusions to models with collapsing bounds and leaky accumulation is important at a conceptual level.

5) Mechanism. Reviewer 3 also requested more details on the proposed mechanism. The idea that changes in a history-dependent drift-rate bias correspond to shifts in endogenous attention from one interpretation to another could be further developed. In particular it seems to imply a more explicit (or at least non-procedural) mechanism for trial-wise changes in drift rate bias. But one could imagine many other mechanisms driving this effect as well (see papers referenced above). As it stands, the connection between shifts in attention and drift-rate bias does seem to be an intuitively plausible explanation (one of many), but a further description of the authors' line of thought would help to convince the reader.

Thank you for this suggestion. We have now elaborated on this idea in the corresponding Discussion paragraph (seventh paragraph), part of which we have quoted above. Our findings indicate that choice history signals bias the accumulation of subsequent evidence. At a neural level, an accumulation may be implemented in at least three ways: through (i) a bias in the neural input (from sensory cortex) to one of the two neural accumulator populations (in association cortex) encoding either choice; (ii) a bias in the way those populations accumulate their inputs (i.e. stronger weights in the connection from input to corresponding accumulator, or reduced leak in that accumulator); or (iii) additional, evidence independent input to one of the accumulators. Scenarios (i) and (ii) precisely match current accounts of the neural implementation of selective attention: a selective boosting of sensory responses to certain features, at the expense of others (Desimone and Duncan, 1995); or a stronger impact of certain sensory responses on their downstream target neurons (e.g. Salinas and Sejnowski, NRN, 2001). And, indeed our besting-fitting accumulator model is in line with scenario (i). Now, the selective amplification of certain sensory responses by top-down attention is commonly explained by selective feedback signals from prefrontal and parietal association cortex to sensory cortex (Desimone and Duncan, 1995) – the same regions that also carry choice history information (Akrami et al., 2018). So, it is tempting to speculate that choice history biases the state of these association circuits, which then feed back these biases to sensory cortex in much the same way as happens during explicit manipulations of attention.

Indeed, while attention is commonly studied in the lab by providing explicit cues, we speculate that an agent’s own choices may be one (among others) important factor that controls the allocation of attention in natural settings, where explicit cues are often not available. Top-down (goal-directed) attention is commonly thought to be accompanied by an awareness of control. We remain agnostic as to whether or not such a sense of control accompanies the spontaneous choice history biases we have studied here. But we feel the functional analogy to attention we have identified here is a potentially important avenue for future research.